

**Influence of anthropogenic emissions and boundary conditions on multi-model**
**simulations of major air pollutants over Europe and North America in the framework**
**of AQMEII3**
Ulas Im[1], Jesper Heile Christensen[1], Camilla Geels[1], Kaj Mantzius Hansen[1], Jørgen Brandt[1],
Efisio Solazzo[2], Ummugulsum Alyuz[3], Alessandra Balzarini[4], Rocio Baro[5,a], Roberto
Bellasio[6], Roberto Bianconi[6], Johannes Bieser[7], Augustin Colette[8], Gabriele Curci[9,10], Aidan
Farrow[11], Johannes Flemming[12], Andrea Fraser[13], Pedro Jimenez-Guerrero[5], Nutthida
Kitwiroon[14], Peng Liu[15], Uarporn Nopmongcol[16], Laura Palacios-Peña[5], Guido Pirovano[4],
Luca Pozzoli[2], Marje Prank[17,18], Rebecca Rose[13], Ranjeet Sokhi[11], Paolo Tuccella[9,10], Alper
Unal[3], Marta G. Vivanco[8,19], Greg Yarwood[16], Christian Hogrefe[20], Stefano Galmarini[2]
[1] Aarhus University, Department of Environmental Science, Frederiksborgvej 399, DK-4000,
Roskilde, Denmark.
[2] European Commission, Joint Research Centre (JRC), Ispra (VA), Italy.
[3] Eurasia Institute of Earth Sciences, Istanbul Technical University, Istanbul, Turkey.
[4] Ricerca sul Sistema Energetico (RSE SpA), Milan, Italy.
[5] University of Murcia, Department of Physics, Physics of the Earth, Campus de Espinardo, Facultad
de Química, 30100 Murcia, Spain.
[6] Enviroware srl, Concorezzo, MB, Italy.
[7] Institute of Coastal Research, Chemistry Transport Modelling Group, Helmholtz-Zentrum
Geesthacht, Germany.
[8] INERIS, Institut National de l'Environnement Industriel et des Risques, Parc Alata, 60550 Verneuil-
en-Halatte, France.
[9] Dept. Physical and Chemical Sciences, University of L'Aquila, L'Aquila, Italy.
[10] Center of Excellence CETEMPS, University of L'Aquila, L'Aquila, Italy.
[11] Centre for Atmospheric and Instrumentation Research (CAIR), University of Hertfordshire,
Hatfield, UK.
[12] European Centre for Medium Range Weather Forecast (ECMWF), Reading, UK.
[13] Ricardo Energy & Environment, Gemini Building, Fermi Avenue, Harwell, Oxon, OX11 0QR, UK.
[14] Environmental Research Group, Kings' College London, London, UK.
[15] NRC Research Associate at Computational Exposure Division, National Exposure Research
Laboratory, Office of Research and Development, United States Environmental Protection Agency,
Research Triangle Park, NC, USA
[16] Ramboll Environ, 773 San Marin Drive, Suite 2115, Novato, CA 94998, USA.
[17] Finnish Meteorological Institute, Atmospheric Composition Research Unit, Helsinki, Finland.
[18] Cornell University, Department of Earth and Atmospheric Sciences, Ithaca, USA.
[19] CIEMAT. Avda. Complutense 40., 28040 Madrid, Spain.
[20] Computational Exposure Division, National Exposure Research Laboratory, Office of Research and
Development, United States Environmental Protection Agency, Research Triangle Park, NC, USA.
[a] now at: Section Environmental Meteorology, Division Customer Service, ZAMG e Zentralanstalt für
Meteorologie und Geodynamik, 1190 Wien, Austria.
**Abstract**
In the framework of the third phase of the Air Quality Model Evaluation International
Initiative (AQMEII3), and as contribution to the second phase of the Hemispheric Transport
of Air Pollution (HTAP2) activities for Europe and North America, the impacts of a 20%
decrease of global and regional anthropogenic emissions on surface air pollutant levels in
2010 are simulated by an international community of regional scale air quality modeling



groups, using different state-of-the-art chemistry and transport models (CTM). The emission
perturbations at the global level, as well as over the HTAP2-defined regions of Europe, North
America and East Asia are first simulated by the global Composition Integrated Forecasting
System (C-IFS) model from European Centre for Medium-Range Weather Forecasts
(ECMWF), which provides boundary conditions to the various regional CTMs participating
in AQMEII3. On top of the perturbed boundary conditions, the regional CTMs used the same
set of perturbed emissions within the regional domain for the different perturbation scenarios
that introduce a 20% reduction of anthropogenic emissions globally as well as over the
HTAP2-defined regions of Europe, North America and East Asia.
Results show that the largest impacts over both domains are simulated in response to the
global emission perturbation, mainly due to the impact of domestic emissions reductions. The
responses of $NO_2$, $SO_2$ and PM concentrations to a 20% percent anthropogenic emission
reductions are almost linear (~20% decrease) within the global perturbation scenario with
however, large differences in the geographical distribution of the effect. $NO_2$, CO and $SO_2$
levels are strongly affected over the emission hot spots. $O_3$ levels generally decrease in all
scenarios by up to ~1% over Europe, with increases over the hot spot regions, in particular in
the Benelux region, by an increase up to ~6% due to the reduced effect of NOx-titration. $O_3$
daily maximum of 8-hour running average decreases in all scenarios over Europe, by up to
~1%. Over the North American domain, the central-to-eastern part and the western coast of
the U.S experience the largest response to emission perturbations. Similar but slightly smaller
responses are found when domestic emissions are reduced. The impact of inter-continental
transport is relatively small over both domains, however, still noticeable particularly close to
the boundaries. The impact is noticeable up to a few percent, for the western parts of the
North American domain in response to the emission reductions over East Asia. $O_3$ daily
maximum of 8-hour running average decreases in all scenarios over North Europe by up to
~5%. Much larger reductions are calculated over North America compared to Europe.
In addition, values of the Response to Extra-Regional Emission Reductions (RERER) metric
have been calculated in order to quantify the differences in the strengths of non-local source
contributions to different species among the different models. We found large RERER values
for $O_3$ (~0.8) over both Europe and North America, indicating a large contribution from non-
local sources, while for other pollutants including particles, low RERER values reflect a
predominant control by local sources.
1. Introduction
Regional air quality modeling has considerably developed during recent decades, driven by
increased concern regarding the impact of air pollution on human health and ecosystems.
Numerous air quality models have been developed by research groups worldwide and are
being widely used for developing and testing emission control policies. Regional atmospheric
chemistry and transport models (CTMs) are widely used to assess the past, present and future
levels of air pollutants from continental to regional scales. There are different sources of
uncertainties in models such as emissions, meteorology, boundary conditions and chemical
schemes that should be taken into account when analyzing results. These uncertainties



become more critical when these models are used for regulatory applications such as impacts
of emission reductions. Multi-model ensembles can help in reducing this uncertainty and
provide a better estimate of impacts under different scenarios (Solazzo et al., 2013; Galmarini
et al., 2013; Kioutsoukis et al., 2017).
Numerous observational and modeling studies show that long-range transport of pollutants
degrade air quality over remote continents (e.g., Wilkening et al., 2000; Holloway et al.,
2003; Akimoto, 2003; Fiore et al., 2009). Although the influence of foreign emissions on
continental scales is seen most frequently in the free troposphere, surface levels can also be
affected, in particular over locations that generally receive clean air masses (e.g. Li et al.,
2002). For example, dust storms and biomass burning can influence the tropospheric
composition on a hemispheric scale (e.g., Husar et al., 2001; Jaffe et al., 2004). Reducing air
pollution levels in surface air would improve public health as exposure to these atmospheric
constituents aggravates respiratory illness and leads to premature mortality (World Health
Organization, 2013; Im et al., 2017; Liang et al., 2017). However, attributing pollution to
specific source regions is complicated due to the different processes influencing
intercontinental transport and by a large hemispheric background and the dominance of local
emissions in contributing to high levels of particular pollutants, such as ozone ($O_3$) (e.g. Fiore
et al., 2009). Given these difficulties, estimates of source-receptor relationships rely heavily
on models.
Stjern et al. (2016), using ten models participating in the second Hemispheric Transport of
Air Pollution (HTAP2) activity, showed that a 20% reduction of global anthropogenic
emissions, leads to significant changes regionally. They found that for North America (NA),
black carbon emissions controls in East Asia are more important than domestic mitigation. In
the framework of the HTAP2 activity, UN (2007) showed that a 20% reduction of North
American NOx emissions leads to a 0.22 ppb decrease in $O_3$ levels over Europe (EU), while a
20% decrease in East Asian NOx emissions leads to a decrease of North American surface $O_3$
levels by 0.12 ppb. The impacts of these emissions changes on the $O_3$ levels in the source
regions are much higher. The impact of lateral boundary conditions (LBC) on concentration
fields simulated by regional-scale air quality models can also be quite significant (Jimenez et
al., 2007; Mathur, 2008; Rudich et al., 2008; Song et al., 2008; Anderrson et al., 2015;
Giordano et al., 2015, Hogrefe et al., 2017; Solazzo et al., 2017a). Recently, Giordano et al.
(2015) showed that the regional models can be very sensitive to the boundary conditions
provided by the global models. Tang et al. (2007) showed that the simulated surface levels
over polluted areas are usually not as sensitive to the variation of LBCs, but are more
sensitive to the magnitude of their background concentrations. Jonson et al. (2017), in the
framework of the HTAP2 activity, showed that for ozone the contributions from the rest of
the world is larger than the effects from European emissions alone, with the largest
contributions from North America and East Asia. The majority of these studies that address
impact of emissions on regional and inter-continental transport employ global models on
coarse spatial resolution or focus on just a few species, such as $O_3$ or carbon monoxide (CO).
On the other hand, studies using regional chemistry and transport models at finer spatial
resolutions mostly focus on sub-regional scales (e.g. Im and Kanakidou, 2012; Huzsar et al.,



2016). Therefore, studies addressing multi-pollutant, source-receptor relationships on inter-
continental and regional scales can provide valuable information on the impact of domestic
and foreign emissions on regional air pollution levels. Multi-model ensembles operating on
fine spatial resolutions can increase accuracy and provide an estimate of uncertainty.
The Air Quality Model Evaluation International Initiative (AQMEII), coordinated jointly by
European Commission, Joint Research Centre (EC-JRC) and the U.S. Environmental
Protection Agency (EPA) has brought together regional chemistry and transport modelling
groups from Europe and North America since 2008 (Rao et al., 2012; Solazzo et al., 2012a,b;
Im et al., 2015 a,b). AQMEII is now running its third phase as a regional sub-project of the
larger Hemispheric Transport of Air Pollution (HTAP), which in turn is a taskforce of Long
Range Transport of Air Pollution program (LTRAP) of United Nations Economic
Commission for Europe (UNECE) (Galmarini et al., 2017). The aim of the study is to assess
the impact of global and HTAP2-defined regional anthropogenic emission reductions of 20%
in Europe, North America and East Asia on major air pollutant levels over Europe and North
America using a multi-model ensemble approach. The study will also investigate the local vs.
non-local contributions to different air pollutant levels, adopting the Response to Extra-
Regional Emission Reductions (RERER) metric developed by the HTAP2 community
(Galmarini et al., 2017).
2. Materials and Methods
In the framework of the AQMEII3 project, fourteen groups contributed to the simulation of
the air pollution levels in Europe (EU) and North America (NA) in the year 2010 (Table 1
and Solazzo et al., 2017b). The emission inventories that are used in the second phase of
AQMEII for Europe and North America (Im et al., 2015a,b) and extensively described in
Pouliot et al. (2015) are also used in AQMEII3. For the EU, the 2009 inventory of MACC
anthropogenic emissions was used. In regions not covered by the Monitoring Atmospheric
Composition & Climate (MACC) inventory, such as North Africa, five modelling systems
have complemented the standard inventory with the HTAPv2.2 datasets (Janssens-Maenhout
et al., 2015). For the NA domain, the 2008 National Emissions Inventory was used as the
basis for the 2010 emissions with 2010-specific adjustments for major point sources, mobile
sources and wildfires (Pouliot et al., 2015). The emissions were then treated with the
SMOKE emissions processing system (Mason et al., 2012). For both continents, the regional
scale emission inventories where embedded in the global scale inventory (Janssens-Maenhout
et al., 2015) to guarantee coherence and harmonization of the information used by the
regional and global scale modelling communities (Galmarini et al., 2017). The majority of the
European groups used MACC emissions over Europe, while FI1 and FRES1 supplemented
the MACC emissions with HTAP emissions over North Africa (Table 1). For NA, the
temporal and vertical allocation of emissions vary between the groups that used the
"SMOKE" files (DE1, US1, US3) and the gridded HTAP files (DK1), however the annual
total mass are exactly the same. Overall, there was a high level of harmonization of emission
inputs even if there were some differences in how they were adapted by each modeling group
for their system. Chemical boundary conditions for both domains were provided by the



European Center for Medium Range Weather Forecasts (ECMWF) Composition – Integrated
Forecast System (C-IFS) model (Flemming et al., 2015)
2.1. Emission perturbations
The perturbation scenarios feature a reduction of 20% of the anthropogenic emissions
globally and in HTAP-defined regions of Europe, North America and East Asia (Table 2).
The choice of 20% was motivated by the consideration that the perturbation would be large
enough to produce a sizeable impact (i.e. more than numerical noise) even at long distances
while small enough to be in the near-linear atmospheric chemistry regime (Galmarini et al.,
2017). The emission reductions are implemented in both the global C-IFS model that
provides the boundary conditions to the participating regional models, as well as in the
regional models. The regional models use the corresponding set of boundary conditions
extracted from the C-IFS model. Among the fourteen groups that participated to the
AQMEII3 base case simulations, twelve groups from Europe and two groups from North
America simulated at least one of the three emission perturbation scenarios, shown in Table
1. Two of the European groups (DE1 and DK1) also simulated the base and the three
perturbation scenarios for the North American domain.
- The global perturbation scenario (GLO) reduces the global anthropogenic emissions
by 20%. This change has been implemented in the C-IFS global model that provides
the boundary conditions to the regional models participating in the AQMEII
ensemble. Therefore, the GLO scenario introduces a change in the boundary
conditions as well as a 20% decrease in the anthropogenic emissions used by the
regional models. Nine groups over the EU domain and four groups over the NA
domain have simulated the GLO scenario.
- The North American perturbation scenario (NAM) reduces the anthropogenic
emissions in North America by 20%. This change has been implemented in the C-IFS
global model that provides the boundary conditions to the regional models used in the
AQMEII ensemble. Therefore, the NAM scenario introduces a change in the
boundary conditions while anthropogenic emissions remain unchanged for Europe,
showing the impact of long-range transport of North American pollutants to Europe
while for North America, the scenario introduces a 20% reduction of anthropogenic
emissions in the HTAP-defined North American region, showing the contribution
from the domestic anthropogenic emissions. Seven groups over the EU domain and
three groups over the NA domain have simulated the NAM scenario.
- The European perturbation scenario (EUR) reduces the anthropogenic emissions in
the HTAP-defined Europe domain by 20%. The EUR scenario introduces a change in
the anthropogenic emissions over the EUR region in the CTMs, showing the
contribution from the domestic anthropogenic emissions. Six groups have simulated
the EUR scenario over the EU domain.
- The East Asian perturbation scenario (EAS) reduces the anthropogenic emissions in
East Asia by 20%. Similar to the NAM scenario for the EU domain, the EAS scenario
introduces a change in the boundary conditions while anthropogenic emissions remain
unchanged in the regional models, showing the impact of long-range transport from



214       East Asia on the NA concentrations. Four groups have simulated the EAS scenario
215       over the NA domain.

In AQMEII, all participating groups were required to upload modelled hourly surface
concentrations to the ENSEMBLE system at EC-JRC, at specified monitoring stations in EU
and NA, as well as surface gridded data (Galmarini et al, 2012; Im et al., 2015a, b; Solazzo et
al., 2017b). This study investigates the impacts of emission perturbations and boundary
conditions on $O_3$, $NO_2$, CO, $SO_2$, $PM_{10}$ and $PM_{2.5}$ levels over Europe and North America.
Differences between each perturbation scenario and the base case (C-IFS global and regional
models run with baseline emissions) are calculated from the gridded hourly pollutant fields,
which are then monthly and annually averaged in order to estimate the impact of the
perturbation of the corresponding emission or boundary condition.
To estimate the contribution of foreign emission perturbations relative to the GLO
perturbation, we have also calculated the RERER metric (Galmarini et al., 2017; Huang et al.,
2017; Jason et al., 2017). For Europe, RERER is calculated using the differences between the
GLO vs BASE as well as the differences between EUR vs. BASE simulations for Europe
(Eq. 1) while for North America; RERER is calculated using the differences between the
GLO vs BASE and NAM vs. BASE simulations (Eq. 2).
$$RERER_{EUR} = \frac{R_{GLO} - R_{EUR}}{R_{GLO}} \qquad\qquad \text{Eq. 1}$$
$$RERER_{NAM} = \frac{R_{GLO} - R_{NAM}}{R_{GLO}} \qquad\qquad \text{Eq. 2}$$
where $R_{GLO}$ is the response of the concentration of a given species to global emission
reduction, $R_{EUR}$ is the response of a concentration of a species to the EUR perturbation for the
European domain, and $R_{NAM}$ is the response of a concentration of a specie to the NAM
perturbation for the North American domain. Therefore, a subset of modelling groups that
have conducted the three simulations (BASE, GLO and EUR/NAM for Europe and North
America, respectively) have been used in the metric calculations (see Table 1). The higher the
local response is, the smaller the RERER metric is. The RERER value can exceed the value 1
when emission reductions lead to increasing concentrations (e.g., $O_3$ titration by nitrogen
monoxide, NO).
**3. Results**
3.1. Model Evaluation
The base case simulation of each model has been evaluated on a monthly basis using
available surface observations from Europe and North America. The observational data used
in this study are the same as the dataset used in the second phase of AQMEII (Im et al.,
2015a,b). The data were provided from the surface air quality monitoring stations operating
in EU and NA. In EU, surface data were provided by the European Monitoring and
Evaluation Programme (EMEP, 2003; http://www.emep.int/) and the European Air Quality
Database (AirBase; http://acm.eionet. europa.eu/databases/airbase/). NA observational data



were obtained from the NAtChem (Canadian National Atmospheric Chemistry) database and
from the Analysis Facility operated by Environment Canada (http://www.ec.gc.ca/natchem/).
The model evaluation results for each model are presented in Fig. 1 and 2, and in Table 3,
along with the results for the multi model (MM) mean and median values. The results show
that the monthly variations of gaseous pollutants are well captured by all models with
correlation coefficients ($r$) generally higher than 0.70. The biases in simulated $O_3$ levels are
generally less than 10% with a few exceptions of up to -35%. The temporal variations of $NO_2$
levels are also well simulated ($r>0.7$), but exhibit much higher biases, with underestimations
up to 75%. CO levels are underestimated by up to 45% while a majority of the models
underestimated $SO_2$ levels by up to 68%. Few models overestimated $SO_2$ by up to 49%. $PM_{10}$
and $PM_{2.5}$ levels are underestimated by 20% to 70%. Slightly higher biases are calculated for
the $PM_{10}$ levels. A more comprehensive evaluation of the models is presented in Solazzo et
al. (2017b), Galmarini et al. (2017) and Im et al. (2017).
C-IFS base case results have also been evaluated along with the regional CTMs, as presented
in Fig. 1 and 2 and in Table 3. The seasonal variations for $O_3$, $NO_2$, CO and $SO_2$ are well
captured with high correlation values of ~0.9. $PM_{10}$ and $PM_{2.5}$ showed a different seasonal
cycle than the observation by not reproducing the wintertime maxima ($r$=~-0.7). C-IFS model
underestimates $O_3$ and CO by ~20% over Europe while $NO_2$ is slightly overestimated
($NMB$=7%). $SO_2$ is overestimated by ~10% over Europe, while $PM_{10}$ and $PM_{2.5}$ levels are
largely underestimated by ~60%, which can be attributed to the lack of secondary aerosol
mechanism in the bulk C-IFS model. Over the North American domain, C-IFS well captures
the seasonal variations of $O_3$, $NO_2$ and CO with correlation coefficients larger than 0.7, while
the seasonal variation of $SO_2$ is not captured by the model ($r$=0.04). The seasonal variations
of $PM_{10}$ and $PM_{2.5}$ are also poorly captured ($r<0.2$). North American $O_3$ levels are slightly
underestimated ($NMB$=-10%), while $NO_2$ and CO are overestimated by ~40% and 20%,
respectively. $SO_2$ is overestimated by 35% 5 while $PM_{10}$ is largely underestimated by ~80
and $PM_{2.5}$ by ~40%. Over both Europe and North America, the wintertime PM levels are
underestimated due to lack of secondary aerosols while the spring summer peaks are
attributed to long range transport of desert dust from the Sahara, which effect mainly the
South East of North America.
3.2. Perturbation Analyses
The annual mean relative differences of each perturbation scenario from the base case
scenario, averaged over all stations, are provided in Table 4 (EU) and Table 5 (NA) for each
modeling group, along with the results for the MM ensemble mean and median. The base
case monthly mean time series for the participating groups are provided in Fig.1 and Fig. 2
for each pollutant, while Fig.3 and Fig. 4 shows the annual mean spatial distribution of the
pollutants from the MM ensemble mean calculations over Europe and North America,
respectively. As seen in the time series figures, there is a large spread among different
groups, owing to the different models used and the different sets of anthropogenic emissions
(Table 1). However, the temporal variation is consistent among all models, in particular for
the gaseous species.



### 3.2.1. Impact of the global emission reduction scenario (GLO)

#### *3.2.1.1. Europe*

The monthly time series of the differences between the GLO and the BASE simulations for each pollutant are presented in Fig. 5. The annual differences are reported in Table 4. Regarding the primary gaseous pollutants, all models simulate the smallest differences during the summer months while the differences are largest in winter. For $O_3$, the simulated differences are positive in winter and negative in summer for all models except for DE1 that simulated a decrease in all months. Results suggest that wintertime $O_3$ over Europe is mainly controlled by anthropogenic emissions. For the other pollutants, results suggest that their levels are mainly controlled by anthropogenic emission throughout the year. The annual difference is smallest for $O_3$, with a reduction of -0.34±1.23 ppb (-1.04±4.00%). The annual mean value of the $O_3$ daily maximum of 8-hour running average decreases by -0.53±1.50 ppb (-1.62±3.99%). $NO_2$ levels decreased by 0.97±0.45 ppb (19.34±1.59%) over Europe while CO levels decreased by 17.35±4.03 ppb (11.22±1.17%), $SO_2$ levels by 0.18±0.05 ppb (20.87±0.93%), $PM_{10}$ by 2.38±0.68 $\mu gm^{-3}$ (15.84±2.12%) and $PM_{2.5}$ by 2.02±0.52 $\mu gm^{-3}$ (18.30±1.75%). Vivanco et al. (2017) found similar reductions regarding the deposition of sulfur and nitrogen species over Europe. Almost all models simulate an overall decrease of annual mean $O_3$ levels over EU (-0.94% to -4.65%), with the exception of TR1 that simulated an increase of 9.31%. Regarding other pollutants, all models simulate a decrease during the simulation period. In general, DE1 and TR1 model groups stand out for introducing the smallest and largest differences, particularly for $O_3$, $NO_2$, and PM.

The geographical distribution of the change in annual mean concentrations in the GLO scenario as simulated by the MM mean is presented in Fig. 6. Regarding $O_3$, most of Europe is characterized by decreased concentrations (Fig.6a). Over central Europe, where most of the primary emissions are located (e.g. NOx), $O_3$ levels slightly increase by ~2%. Emission hotspots, in particular the Benelux area stands out with largest increases (~6%) due to decreased NOx-titration effect, which can also be seen in Fig. 6b. In addition, $O_3$ levels over the northern parts of Germany and France, and southern UK are increasing in response to emission reductions. There is also a clear decrease in CO levels (Fig.6c), in particular over central Europe by up to ~16%. All primary species decrease over the whole domain, especially over the industrial hot spots such as in Poland, Po Valley and the Benelux area (Fig.6d). PM levels decrease throughout the domain by up to ~20% (Fig.6e and f).

#### *3.2.1.2. North America*

The seasonal variation of the impact of 20%-decreased global emissions on the North American pollutant levels are presented in Fig.7. All models simulated a small decrease of 3% to 5% (Table 5) in $O_3$ levels with the largest differences in spring to summer (Fig.7a). The mean response to the emission perturbation is estimated to be -1.39 ± 0.27 ppb (-3.52 ± 0.80%). The annual mean value of the $O_3$ daily maximum of 8-hour running average decreases by -1.93±0.14 ppb (-4.51±0.45%). All models simulated a largest $NO_2$ response in winter. Most models simulated a decrease of $NO_2$ levels while DK1 estimated an increase


(Fig.7b). As shown is Table 5, the models simulated a $NO_2$ response of ~0.4 – 1.2 ppb (-17.8
± 0.78%). Regarding CO, all models simulated very clear seasonal profile of the response to
emission reductions, with maximum change in late winter/early spring and the minimum
change in summer. Most models simulated a change around -15 to -25 ppb (~11%); with the
exception of the DE1 model simulating a decrease of ~9 ppb (~7.9%). The MM mean
response is calculated to be 19.2 ± 6.9 ppb (-11 ± 2.3%). The impact of the emission
reduction on $SO_2$ levels was calculated to be -0.25 ppb to -0.48 ppb (-20.3 ± 0.2%).
The response of $PM_{10}$ levels to the global emission reduction was calculated to be -2.4 ± 1.8
$\mu gm^{-3}$ (-32.1 ± 26.6%) (Table 5). The largest relative change was calculated for DE1 (~63%).
DK1 has almost a flat response around -1 $\mu gm^{-3}$, while DE1, which is overlapped with the
Median line, and US3 have maximum responses in early spring and mid-autumn, while they
simulate a minimum response in winter and late spring. Regarding $PM_{2.5}$, the multi-model
mean response was calculated to be -1.5 ± 0.9 $\mu gm^{-3}$ (-17.2 ± 1.8%). DK1 (overlapped with
the Median) and US3 simulated the minimum response in May (Fig.7f), while US3 has a
slightly higher second minimum in September. This minimum is also simulated by DE1 as
the minimum response. DE1 simulates the lowest response among the three models.
The spatial distributions of response of different pollutants to the GLO scenario are presented
in Fig.8. $O_3$ levels are reduced over most of the domain (Fig.8a), with slight increases over
the emission hotspots due to reduced effect of NOx-titration, as seen in Fig.8b, as well as
decreased CO levels over the whole domain (Fig.8c). $SO_2$ levels are also decreased
throughout the domain (Fig.8d), with the largest reductions over the Atlantic (attributable to
reduction in shipping emissions). The western part of the continent is characterized by the
lowest reductions. PM levels are reduced throughout the domain by up to 25% (Fig.8e and f),
with the largest reductions over the eastern and central parts of the domain. A large decrease,
more pronounced in the $PM_{2.5}$ response, can also be seen over California in the western
coastal United States.
3.2.2. Impact of the North American emission reduction scenario (NAM)
*3.2.2.1. Europe*
NA emission reductions account for a reduction of European $O_3$ levels of -0. 22±0.07 ppb (-
0.75±0.14%), with all models simulating a decrease of -0.51% to 0.86%, except for the ES1
model that simulated an increase of 1.31% (Table 4). This decrease is in agreement with
previous studies, such as the HTAP2 study (UN, 2017) that calculated an $O_3$ reduction over
Europe of 0.22 ppb in response to a 20% decrease in the North American NOx emissions, and
Fiore et al. (2009) that simulated a MM mean response of -0.4 ppb in response to a 20%
reduction of anthropogenic emissions in North America. $NO_2$ levels increase slightly by
0.16±0.01%. The annual mean value of the $O_3$ daily maximum of 8-hour running average
decreases by -0.15±0.27 ppb (-0.45±0.77%). CO levels also decreased over the EU domain
by -1.39±0.27 ppb (-0.96±0.22%), much higher than ~0.1 ppb calculated by Fiore et al.
(2009). $PM_{10}$ and $PM_{2.5}$ levels also decreased slightly by -0.03±0.03 $\mu gm^{-3}$ (-0.21±0.7%) and
-0.02± 0.02 $\mu gm^{-3}$ (-0.18±0.25%), respectively. The models had different $SO_2$ responses to



the NA emissions. Overall, DE1, ES1 and FRES1 simulated almost no change in the surface
SO$_2$ levels while DK1, ES1 and TR1 simulated an increase (0.10%, 5.75% and 0.01%,
respectively) and FI1 and UK1 simulated a decrease (-0.02% and -0.03%, respectively).
Different responses can be due to different model setups including aqueous chemistry,
vertical resolutions and aerosol modules (Solazzo et al., 2017).
All models were consistent in simulating the largest impact on O$_3$ during spring and a second
lower peak in autumn (Fig.9a). Surface mean NO$_2$ concentrations (Fig.9b) increased in most
models except for FRES1 that simulated a small decrease except for winter. FI1 also
simulated a decrease during the winter period extending to the transition periods. All models,
except for ES1, simulated a similar response of CO concentrations to perturbation to NA
emissions, with a distinct seasonality (Fig.9c). The SO$_2$ response in models is also consistent
except for the winter period where there is a large spread in magnitude and the sign of the
response (Fig.9d).
O$_3$ levels decreased slightly over the entire European domain by up to 3% (Fig.10a). The
largest impact is simulated over the western boundary and gradually decreases eastwards.
The response of O$_3$ levels to NAM emissions is more evident during spring where there is a
clear transport from Atlantic to the western/northwestern parts of Europe such as the U.K,
northern France and Scandinavia (Fig. S2a). The transport of Atlantic air masses is also
shown for the springtime CO levels over Europe (Fig. S2a). The ensemble mean simulates a
slight increase of up to 3% in NO$_2$ levels over Europe (Fig.10b). Along with the O$_3$ levels,
CO levels show the largest decrease over northwestern Europe by up to ~2%. SO$_2$ levels
increased over the whole domain, in particular over Eastern Europe and the Alpine region
(Fig.10d), due to a decrease in the oxidative capacity of the atmosphere (see Fig.10a for O$_3$),
leading to a decrease in the SO$_2$ to SO$_4$ conversion. This results in an increase of the SO$_2$
levels and a decrease in the PM$_{2.5}$ levels (Fig.10e and f).
*3.2.2.2. North America*
The response of North American pollutant levels to a 20% reduction of North American
anthropogenic emissions (implemented in both C-IFS and the regional CTMs) are presented
in Table 5. The NAM scenario led to a decrease of annual mean O$_3$ levels over North
America by -0.36 ppb (US3) to -0.92 ppb (DE1), with *MM* ensemble mean calculated to be -
0.65±0.28 ppb (-1.45±0.88%), in agreement with Fiore et al. (2009) that calculated a decrease
of ~1 ppb. The annual mean value of the O$_3$ daily maximum of 8-hour running average
decreases by -1.11±0.11 ppb (-2.60±0.36%), very similar to the change over Europe.
Consequently, the largest change in NO$_2$ levels were simulated by US3 (-1.17 ppb) and
smallest by DE1 (-0.36 ppb). The MM mean response of NO$_2$ is calculated to be -0.71±0.41
ppb (-17.24±0.58%). Similar to NO$_2$, the largest response in CO levels were simulated by
US3 (-19.87 ppb) and the smallest by DE1 (-3.84 ppb), leading to a MM mean response of -
12.35±8.06 ppb (-7.01±3.60%). As seen in Table 5, DE1 simulated a much lower absolute
and relative change in CO response compared to DK1 and US3. SO$_2$ levels decreased by -
0.32 ppb to -0.48 ppb, leading to a MM mean response of -0.37±0.09 ppb (-20±0.12%). PM$_{10}$
levels decreased -1.78±2.08 μgm$^{-3}$ (-15.78±3.26%). As seen in Table 5, DK1, simulated a



very low response to the NAM scenario, by ~0.60 $\mu gm^{-3}$, compared to the DE1 and the US3
groups that simulated a $PM_{10}$ response of -2.02 $\mu gm^{-3}$ and -4.19 $\mu gm^{-3}$, respectively.
However, the relative responses are not very different between the different groups (~16%).
The response of $O_3$ to the NAM scenario is largest in summer (Fig.11a): June for DK1 and
US3 and August for DE1. The $O_3$ response clearly shows a difference from the GLO
response in spring, suggesting the impact of long-range transport in spring that does not
appear in the perturbation of the local emissions only. The largest $NO_2$ response (Fig.11b) is
simulated by US3, similar to the response to the GLO scenario. The response of CO to the
reductions in local emissions (Fig.11c) is different from the response to the global reduction,
where DK1 and US3 has the minimum response in spring and DE1 has the minimum
response in autumn. The response of $SO_2$ and PM to GLO and NAM are similar, suggesting
the main drivers of $SO_2$ and PM levels are local emissions.
Annual mean $O_3$ levels show large reductions (~20%) over the eastern parts of the domain,
while there are slight increases or less pronounced decreases over the western parts of the
domain (Fig.12a), associated with larger NOx reductions (Fig.12b). CO and $SO_2$ levels are
mostly reduced over the central to eastern parts of the domain (Fig.12c and d, respectively),
with shipping impacts over the Atlantic being more pronounced on $SO_2$ levels. The western
parts of the U.S. experiences smaller $SO_2$ reductions (~5-10%) and slight increases over the
southwestern U.S. The response of PM to the NAM scenario (Fig.12e and f) is very similar to
the response to the GLO scenario (Fig.8e and f).
3.2.3. Impact of the European emission reduction scenario (EUR)
$O_3$ levels increase slightly by 0.01±0.40 ppb (0.25±1.35%) in response to the 20% reduction
of the anthropogenic emissions from Europe (Table 4). This response is much lower than
Fiore et al. (2009) that calculated a MM mean response of 0.8 ppb. However, as seen in
Fig.13a, the positive mean response together with the large standard deviation is due to the
DE1 model that simulated a decrease (-2.33%), while other groups simulated an increase
(0.39% to 1.72%). There is a distinct seasonality in the response with winter levels increasing
with reduced emissions and summer levels decreasing, following the emission temporal
variability. The annual mean value of the $O_3$ daily maximum of 8-hour running average
decreases by -0.21±0.10 ppb (-0.62 0.24%). $NO_2$ concentrations decreased by -0.75±0.26
(17.68±0.90%), with a similar seasonal response of $SO_2$ levels (-17.52±1.70%) and CO levels
(-6.26±1.07%), consistent with the findings of Vivanco et al. (2017). An opposite seasonal
variation is calculated for the $O_3$ response (Fig. 13.b-d)., The DE1 model also stands out in
the $NO_2$ response together with the FRES1 model in the magnitude of the response (Fig.13b).
$PM_{10}$ and $PM_{2.5}$ levels have similar responses to the emissions reduction (-14.43±2.84% and -
15.67±2.12%, respectively) with similar seasonality.
The MM mean geographical distribution of the $O_3$ response is very similar with that of the
GLO perturbation (Fig.14a), with relatively smaller decreases by up to ~3%. $O_3$ levels
increase over the central and in particular over northwestern Europe by up to ~6%. $NO_2$
levels decrease uniformly over the entire domain by up to ~20% (Fig.14b). CO levels



decrease over the emission sources, mainly over central and Eastern Europe (Fig.14c). PM
levels also decrease over the entire domain, especially over central and Eastern Europe
(Fig.14e and f).
3.2.4. Impact of the East Asian emission reduction scenario (EAS)
As seen in Table 5, the impacts of East Asian emissions on North American $O_3$ levels are
much lower than the impacts from the reductions in global and local emissions. The largest
impact is simulated by DE1 as -0.99 ppb (-0.35%), while other models give similar responses
(~0.60 ppb; -0.20%). The $O_3$ response as calculated by the MM mean ensemble is -0.25±0.07
ppb, in agreement with the HTAP2 findings and Fiore et al. (2009). The annual mean value
of the $O_3$ daily maximum of 8-hour running average decreases by -0.28±0.07 ppb (-
0.65±0.20%). $NO_2$ and $SO_2$ response to reductions in EAS emissions were simulated to be
very small (-0.04±0.08% and 0.01±0.02%, respectively). The CO response to EAS was
simulated to be -2.60 ppb (DE1) to -4.16 ppb (DK1), with the MM mean response of -
3.37±0.68 ppb (-2±0.29%). Regarding $PM_{10}$, DE1 simulated a very large response (~-0.56
$\mu gm^{-3}$) compared to DK1 and US3 (~-0.05 $\mu gm^{-3}$), leading to a MM mean response of -
0.21±0.30 $\mu gm^{-3}$ (-5.63±8.50%). However, the $PM_{2.5}$ response was much lower (-0.02±0.03
$\mu gm^{-3}$; -0.20±0.35%), suggesting that the $PM_{2.5}$ levels are largely driven by local emissions.
The $O_3$ response to EAS emission reductions was highest in spring and autumn, suggesting
that long-range transport is important in these seasons (Fig.15a). The $NO_2$ response was
negative, being maximum in winter and minimum in summer, except for DK1 showing an
increase in $NO_2$ levels in all seasons (Fig.15b). The impact of EAS emissions on North
American CO levels showed a distinct seasonality (Fig.15c), similar to the impact of the
global emission reductions (Fig.5c), suggesting that regional CO levels over North America
are driven by both local emissions and long-range transport. The response of $SO_2$ to East
Asian emission reductions varied largely from model to model with US3 showing an overall
reduction while DE1 and DK1 simulated increases in winter, spring, and autumn, and
decreases in summer (Fig.15d). The $PM_{10}$ response simulated by DK1 (overlapped with the
median) and US3 were simulated to be small, being largest in spring (Fig.15e). However,
DE1 simulated a large and opposite response, with spring having the smallest response and
winter with the largest response. DE1 also simulated a different $PM_{2.5}$ response in terms of
the sign of the change and thus, seasonality in response to DK1 and US3 (Fig.15f). Largest
differences were simulated in spring, similar to $PM_{10}$ by DK1 and US3, while DE1 simulated
the largest response in winter and summer and the spring response was minimum.
The impact of the East Asian emissions over the western parts of North America is clearly
seen for all pollutants in Fig.16. The impacts are low for all pollutants, being up to 5%. The
impacts are particularly pronounced for CO (Fig.16c), $SO_2$ (Fig.16d) and PM (Fig.16e and f).
The largest $O_3$ response was simulated over the northwestern parts of North America
(Fig.16a). The springtime transport of O3 from East Asia is more evident compared to the
annual average of the perturbation response (Fig. S3a), where the western NA $O_3$ levels
decrease by up to ~1.5%. The springtime CO levels also decrease by up to 6% (Fig. S3b),
showing the importance of long-range transport from East Asia.



### 3.2.5. RERER analyses

As discussed in Section 2, the RERER metric (Galmarini et al., 2017; Hang et al., 2017; Jason et al., 2017) is designed to quantify the relative impact of local vs. non-local emission sources on pollutant levels in the receptor regions EU and NA. Using gridded hourly pollutant concentrations from the base case, GLO and EUR simulations, the RERER metrics for the EU have been calculated for the annual mean concentrations response for the individual groups as well as for the ensemble mean. For the NA domain. The RERER metrics have been calculated using the base case, GLO and NAM simulations. Table 6 presents the RERER metric calculated for the European domain. The table shows differences in the strengths of non-local source contributions to different species among the different models. Regarding the RERER metric for $O_3$ in Europe, most values calculated are below one, except for the IT1 model, which shows a significant increase of $O_3$ levels in Europe in response to emission reductions compared with the other models. A RERER value of 0.8-0.9 is calculated for the majority of models, implying the dominance of non-local sources in Europe, except for the DE1 model, where the RERER value is lower (~0.5), giving an equal contribution of local vs. non-local sources in Europe. The MM mean RERER value for $O_3$ is ~0.8, showing a much larger contribution of non-local sources compared to local sources in Europe. This result is in agreement with, however slightly smaller, Jonson et al. (2017) that calculated a MM mean RERER value of 0.89.

Regarding $NO_2$, the RERER metrics (< 0.4) show that $NO_2$ is controlled by local sources. In addition, the RERER metrics calculated for DE1 and FI1 are slightly negative, implying that the signal is not sensitive to non-local emissions. RERER calculated for the ensemble mean for $NO_2$ (~0.2) also shows the high sensitivity of $NO_2$ concentrations to local sources. The RERER metric calculations for CO shows similar contributions from local vs. non-local sources, with RERER values of 0.4-06, except for IT1. IT1 has a RERER metric value of ~0.9 suggesting a large contribution of non-local sources, leading to the higher sensitivity of CO to non-local sources compared to other model groups. The RERER values calculated for the ensemble mean (~0.6) shows a slightly larger contribution of non-local sources compared to local sources. The MM mean RERER value of 0.55 for CO from this study is in very good agreement with Jonson et al. (2017) that calculated a MM mean RERER of 0.51. RERER metrics calculated for $SO_2$ are also in the low range (0-0.4). While DE1 and FI1 show almost no signal for the non-local contribution, DK1, IT1 and UK1 are in the higher end of the range. The CO MM mean RERER value of ~0.3 shows that CO levels are largely controlled by local emissions. Finally, the metrics calculated for $PM_{10}$ and $PM_{2.5}$ shows that local sources are the main contributor to the PM levels in Europe (RERER = ~0 - 0.3), leading to an ensemble mean contribution of local sources (RERER = ~0.2).

Regarding the local vs. non-local contributions to different pollutants over the North American domain, three groups out of four simulated the GLO and NAM scenarios needed to calculate the RERER metrics. RERER metrics show that $O_3$ is largely controlled by non-local sources. European model groups DE1 and DK1 simulate a larger influence of non-local sources (~0.8 - ~0.9) compared to the US3 group, which simulated lower RERER metric values of ~0.5, indicating that $O_3$ levels are driven equally by local and non-local sources.





This lower value is also consistent with the findings of Huang et al. (2017), who simulated
the largest impacts on $O_3$ in May and June with RERER values around ~0.5. The ensemble
mean shows that $O_3$ responses are largely attributable to non-local sources (RERER = ~0.8),
which are similar to those found for Europe. RERER metric values calculated for $NO_2$ by
different models (RERER = ~0 – ~0.2) and the ensemble mean (RERER = 0.05) clearly
shows that $NO_2$ is controlled by local sources, similar to the Europe case. The sensitivity of
CO to local and non-local sources are similar to those for $O_3$, with DE1 and DK1 simulating a
large contribution from non-local sources while US1 shows that CO is controlled equally by
local and non-local sources (RERER = 0.5). Similar to $NO_2$, all models show that $SO_2$ is
largely driven by local sources with RERER values between ~0.1 and ~0.2. Regarding the
particles, models simulate very similar responses to changes in the local and non-local
sources. RERER values are calculated to be ~0.08 and ~0.11 for $PM_{10}$ and $PM_{2.5}$,
respectively, showing the large local contribution compared to non-local sources.

**CONCLUSIONS**
In the framework of the third phase of the Air Quality Model Evaluation International
Initiative (AQMEII3), the impacts of local vs. foreign emissions over the European and North
American receptor regions are simulated by introducing a 20% decrease of global and
regional emissions by research groups, using different state-of-the-art chemistry and transport
models. The emission perturbations were introduced globally, as well as over the HTAP2-
defined regions of Europe, North America and East Asia. Base case and the perturbation
scenarios are first simulated using the global C-IFS global model, which provides the
boundary conditions to the regional CTMs.
The base case simulation of each model has been evaluated against surface observations from
Europe and North America. The temporal variabilities of all pollutants are well captured by
all models with correlations generally higher than 0.70. $O_3$ levels are generally simulated
with a *MNB* less than 10% with few exceptions of *MNB* values up to -35%. $NO_2$, CO and
SO2 levels are simulated with underestimations up to 75%, 45% and 68%, respectively. $PM_{10}$
and $PM_{2.5}$ levels are underestimated by 20% to 70%, with slightly higher biases in $PM_{10}$
levels.
Results from the perturbation simulations show that the largest impacts over both Europe and
North American domains are simulated in response to the global emission perturbation
(GLO). These responses are similar, however slightly lower, as compared to the local
emission perturbation scenarios for Europe (EUR) and North America (NAM). In contrast to
the GLO scenario, $O_3$ levels over Europe slightly increase by 0.13 ppb (0.02%). The annual
mean value of the $O_3$ daily maximum of 8-hour running average decreases in all scenarios
over Europe, highest in the GLO scenario by ~1% and lowest in the NAM scenario by
~0.3%. Over North America, the annual mean value of the $O_3$ daily maximum of 8-hour
running average decreased by ~5% in the GLO scenario, 3% in the NAM scenario and 0.7%
in the EAS scenario. The impact of foreign emissions simulated by the NAM scenario for



Europe and EAS scenario for North America were found to be lowest, however still
noticeable, particularly close to the boundaries. This impact is especially noticeable (up to
only a few percent) for the western parts of the North American domain in response to the
emission reductions over East Asia. The response is almost linear (~20% decrease) to the
change in emissions for $NO_2$, $SO_2$ and PM in the global perturbation scenario (GLO), while
$O_3$ levels decrease slightly (~1%).
Despite these small differences, there are large geographical differences. $NO_2$, CO and $SO_2$
levels are mainly affected over emission hot spots in the GLO scenario as well as in the EUR
scenario for Europe and the NAM scenario for North America. $O_3$ levels increase over the hot
spot regions, in particular the Benelux region in Europe, by up to ~6% due to the reduced
effect of NOx-titration. Over the North American domain, the central-to-eastern part and the
western coast of the U.S experience the largest response to the global emission perturbation.
For most of the pollutants, there is distinct seasonality in the responses particularly to the
global and local emission perturbations. The largest responses are calculated during winter
months, where anthropogenic emission are highest, except for $O_3$, where largest responses are
seen during spring/summer months, suggesting photochemistry still plays an important role in
$O_3$ levels.
The RERER metrics have been calculated to examine the differences in the strengths of non-
local source contributions to different species among the different models. The large RERER
values over Europe and North America for $O_3$ (~0.8), show a larger contribution of non-local
sources, while for other gaseous pollutants ($NO_2$, CO and $SO_2$) and particles ($PM_{10}$ and
$PM_{2.5}$), low RERER values (< 0.5) indicate that these pollutants are largely controlled by
local sources. Results show that the contribution of local sources on $NO_2$, $SO_2$ and PM levels
are larger in North America compared to Europe, while for CO, local sources have a larger
share in Europe in comparison with North America.
Overall results show that there is a large spread among the models, although the majority of
the models simulate a similar seasonal variation. These differences suggest that despite the
harmonization of inputs, such as emissions and boundary conditions, to regional models,
there are still large differences between models, such as different gas phase and aerosol
modules, deposition schemes, meteorological drivers and spatial and vertical resolutions.
Therefore, the use of multi model ensembles can help to reduce the uncertainties inherent in
individual models.

**ACKNOWLEDGEMENTS**
We gratefully acknowledge the contribution of various groups to the third air Quality Model
Evaluation international Initiative (AQMEII) activity. Joint Research Center Ispra/Institute
for Environment and Sustainability provided its ENSEMBLE system for model output
harmonization and analyses and evaluation. The views expressed in this article are those of
the authors and do not necessarily represent the views or policies of the U.S. Environmental
Protection Agency. Aarhus University gratefully acknowledges the NordicWelfAir project



funded by the NordForsk's Nordic Programme on Health and Welfare (grant agreement no.
75007), the REEEM project funded by the H2020-LCE Research and Innovation Action
(grant agreement no.: 691739), and the Danish Centre for Environment and Energy (AU-
DCE). University of L'Aquila thanks the EuroMediterranean Center for Climate Research
(CMCC) for providing the computational resources. RSE contribution to this work has been
financed by the research fund for the Italian Electrical System under the contract agreement
between RSE S.p.A. and the Ministry of Economic Development – General Directorate for
Nuclear Energy, Renewable Energy and Energy Efficiency in compliance with the decree of
8 March 2006. CIEMAT has been financed by the Spanish Ministry of Agriculture and Food,
Fishing and Environment. University of Murcia thanks the Spanish Ministry of Economy for
the research contract CGL2014-59677-R (also partially funded by the FEDER programme).

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

Table 1. Key features (meteorological/chemistry and transport models, emissions, horizontal and vertical grids) of the regional models participating to the AQMEII3 health impact study and the perturbation scenarios they performed.

| Group Code | Model | Emissions[1] | Horizontal Resolution | Vertical Resolution | Europe | | | | North America | | | |
|---|---|---|---|---|---|---|---|---|---|---|---|---|
| | | | | | BASE | GLO | NAM | EUR | BASE | GLO | EAS | NAM |
| DE1 | COSMO-CLM/CMAQ | HTAP | 24 km × 24 km | 30 layers, 50 hPa | × | × | × | × | × | × | × | × |
| DK1 | WRF/DEHM | HTAP | 50 km × 50 km | 29 layers, 100 hPa | × | × | × | × | × | × | × | × |
| ES1 | WRF/CHEM | MACC | 23 km × 23 km | 33 layers, 50 hPa | × | | × | | | | | |
| FI1 | ECMWF/SILAM | MACC+HTAP | 0.25° × 0.25° | 12 layers, 13 km | × | × | × | × | | | | |
| FRES1 | ECMWF/CHIMERE | MACC+HTAP | 0.25° × 0.25° | 9 layers, 50 hPa | × | × | × | × | | | | |
| IT1 | WRF/CAMx | MACC | 23 km × 23 km | 33 layers, 50 hPa | × | × | | × | | | | |
| IT2 | WRF/CHEM | MACC | 23 km × 23 km | 14 layers, 8 km | × | × | | | | | | |
| NL1 | LOTOS/EUROS | MACC | 0.50° × 0.25° | 4 layers, 3.5 km | × | | | | | | | |
| TR1 | WRF/CMAQ | MACC | 30 km × 30 km | 24 layers, 10hPa | × | × | × | | | | | |
| UK1 | WRF/CMAQ | MACC | 15 km × 15 km | 23 layers, 100 hPa | × | × | × | × | | | | |
| UK2 | WRF/CMAQ | HTAP | 30 km × 30 km | 23 layers, 100 hPa | × | × | × | | | | | |
| UK3 | WRF/CMAQ | MACC | 18 km × 18 km | 35 layers, 16 km | × | × | × | | | | | |
| US1 | WRF/CAMx | SMOKE | 12 km × 12 km | 29 layers, 97.5 hPa | | | | | × | × | × | |
| US3 | WRF/CMAQ | SMOKE | 12 km × 12 km | 35 layers, 50 hPa | | | | | × | × | × | × |

[1] MACC: Modelling group used only the MACC emissions, MACC+HTAP: Modelling group used MACC emissions for Europe and HTAP
emissions over North Africa.





Table 2. Perturbations of global/regional anthropogenic emissions and boundary conditions in the perturbation scenarios.

| | GLO | Europe | | North America | |
|---|---|---|---|---|---|
| | | NAM | EUR | NAM | EAS |
| Emissions | -20% | - | -20% | -20% | - |
| Boundary conditions (Emissions in the IFS model) | -20% | -20% | -20% | -20% | -20% |







Table 3. Monthly statistics of Pearson's Correlation ($r$), Normalized Mean Bias (*NMB*: %), Normalized Mean Gross Error (*NMGE*: %) and Root
Mean Square Error (*RMSE*: µg m$^{-3}$ for Europe, while ppb for gases and µg m$^{-3}$ for particles for North America) calculated for each model group.

| | | EUROPE | | | | | | | | | | | | | NORTH AMERICA | | | | | | |
| --- | --- | --- | --- | --- | --- | --- | --- | --- | --- | --- | --- | --- | --- | --- | --- | --- | --- | --- | --- | --- | --- |
| | | DE1 | DK1 | ES1 | FI1 | FRES1 | IT1 | IT2 | TR1 | UK1 | UK2 | MEAN | MEDIAN | C-IFS | DE1 | DK1 | US1 | US3 | MEAN | MEDIAN | C-IFS |
| O₃ | r | 0.63 | 0.90 | 0.82 | 0.83 | 0.91 | 0.92 | 0.93 | 0.87 | 0.92 | 0.90 | 0.93 | 0.92 | 0.89 | 0.78 | 0.59 | 0.89 | 0.87 | 0.84 | 0.83 | 0.71 |
| | NMB | 0.10 | 0.07 | -0.14 | -0.36 | -0.10 | 0.04 | -0.14 | 0.09 | 0.08 | -0.03 | -0.04 | -0.04 | -0.20 | 0.12 | 0.22 | 0.14 | -0.02 | 0.09 | 0.11 | -0.10 |
| | NMGE | 0.17 | 0.12 | 0.15 | 0.36 | 0.12 | 0.13 | 0.15 | 0.26 | 0.11 | 0.09 | 0.08 | 0.08 | 0.20 | 0.17 | 0.23 | 0.14 | 0.08 | 0.12 | 0.13 | 0.19 |
| | RMSE | 12.68 | 8.81 | 11.58 | 23.13 | 9.01 | 8.54 | 10.94 | 17.66 | 8.05 | 6.79 | 5.91 | 6.31 | 14.63 | 6.16 | 9.81 | 5.72 | 3.23 | 4.63 | 5.28 | 7.31 |
| NO₂ | r | 0.80 | 0.88 | 0.89 | 0.95 | 0.74 | 0.90 | 0.92 | 0.90 | 0.85 | 0.85 | 0.95 | 0.93 | 0.92 | 0.99 | 0.92 | 0.94 | 0.93 | 0.98 | 0.99 | 0.91 |
| | NMB | -0.75 | -0.38 | -0.47 | 0.00 | 0.05 | -0.29 | -0.30 | 0.58 | -0.32 | -0.06 | -0.17 | -0.24 | 0.07 | -0.18 | -0.35 | 0.05 | 0.31 | -0.03 | -0.02 | 0.41 |
| | NMGE | 0.75 | 0.38 | 0.47 | 0.20 | 0.23 | 0.29 | 0.30 | 0.58 | 0.32 | 0.17 | 0.18 | 0.24 | 0.20 | 0.18 | 0.35 | 0.10 | 0.31 | 0.06 | 0.02 | 0.41 |
| | RMSE | 9.38 | 5.41 | 6.00 | 2.89 | 3.44 | 4.43 | 4.15 | 7.39 | 4.65 | 2.74 | 2.70 | 3.49 | 2.59 | 1.01 | 2.05 | 0.62 | 1.77 | 0.40 | 0.26 | 2.30 |
| CO | r | 0.83 | 0.76 | 0.74 | 0.88 | 0.82 | 0.84 | 0.79 | 0.87 | 0.63 | 0.72 | 0.92 | 0.84 | 0.91 | 0.79 | 0.74 | 0.74 | 0.73 | 0.88 | 0.82 | 0.80 |
| | NMB | -0.42 | -0.42 | -0.44 | -0.27 | -0.32 | -0.38 | -0.44 | -0.20 | -0.41 | -0.43 | -0.33 | -0.38 | -0.25 | -0.19 | -0.07 | -0.06 | -0.04 | -0.07 | -0.07 | 0.17 |
| | NMGE | 0.42 | 0.42 | 0.44 | 0.27 | 0.32 | 0.38 | 0.44 | 0.21 | 0.41 | 0.43 | 0.33 | 0.38 | 0.25 | 0.19 | 0.11 | 0.08 | 0.08 | 0.08 | 0.07 | 0.17 |
| | RMSE | 128.62 | 134.31 | 132.78 | 89.99 | 107.81 | 128.14 | 135.83 | 70.04 | 130.21 | 135.82 | 106.98 | 123.61 | 84.73 | 40.27 | 24.90 | 22.44 | 20.51 | 19.94 | 20.41 | 37.30 |
| SO₂ | r | 0.85 | 0.90 | 0.88 | 0.86 | 0.87 | 0.86 | 0.86 | 0.54 | 0.83 | 0.83 | 0.93 | 0.92 | 0.70 | 0.79 | 0.81 | 0.80 | 0.78 | 0.87 | 0.78 | 0.70 |
| | NMB | -0.01 | -0.47 | -0.65 | -0.20 | -0.16 | -0.30 | -0.55 | 0.04 | -0.13 | 0.20 | -0.19 | -0.10 | 0.41 | -0.46 | -0.42 | 0.07 | -0.13 | -0.19 | -0.13 | -0.59 |
| | NMGE | 0.24 | 0.48 | 0.65 | 0.28 | 0.22 | 0.31 | 0.55 | 0.28 | 0.19 | 0.28 | 0.21 | 0.12 | 0.45 | 0.46 | 0.42 | 0.11 | 0.13 | 0.19 | 0.13 | 0.59 |
| | RMSE | 0.92 | 1.47 | 2.03 | 0.95 | 0.80 | 1.23 | 1.71 | 1.14 | 0.86 | 1.05 | 0.76 | 0.58 | 1.39 | 1.27 | 1.18 | 0.32 | 0.40 | 0.53 | 0.40 | 1.02 |
| PM₁₀ | r | 0.86 | 0.82 | 0.17 | 0.41 | 0.82 | 0.60 | 0.10 | 0.52 | 0.71 | 0.71 | 0.87 | 0.73 | 0.70 | -0.31 | -0.47 | NA | 0.07 | 0.47 | -0.07 | 0.02 |
| | NMB | -0.71 | -0.59 | -0.47 | -0.42 | -0.51 | -0.20 | -0.48 | -0.25 | -0.47 | -0.42 | -0.41 | -0.45 | -0.62 | -0.67 | -0.84 | NA | -0.25 | -0.44 | -0.46 | -0.86 |
| | NMGE | 0.71 | 0.59 | 0.47 | 0.42 | 0.51 | 0.25 | 0.48 | 0.26 | 0.47 | 0.42 | 0.41 | 0.45 | 0.62 | 0.67 | 0.84 | NA | 0.27 | 0.44 | 0.46 | 0.86 |
| | RMSE | 20.43 | 18.25 | 16.16 | 14.67 | 15.74 | 9.78 | 16.48 | 10.45 | 14.78 | 13.72 | 13.15 | 14.63 | 19.87 | 20.42 | 25.09 | NA | 9.85 | 13.51 | 14.74 | 25.58 |
| PM₂.₅ | r | 0.89 | 0.86 | 0.24 | 0.58 | 0.84 | 0.75 | 0.11 | 0.62 | 0.77 | 0.77 | 0.89 | 0.82 | -0.73 | 0.52 | 0.02 | NA | 0.54 | 0.61 | 0.56 | 0.18 |
| | NMB | -0.64 | -0.47 | -0.27 | -0.27 | -0.36 | -0.19 | -0.48 | -0.17 | -0.40 | -0.28 | -0.32 | -0.33 | -0.59 | -0.63 | -0.14 | NA | 0.17 | -0.15 | -0.08 | -0.39 |
| | NMGE | 0.64 | 0.47 | 0.35 | 0.30 | 0.36 | 0.24 | 0.49 | 0.24 | 0.41 | 0.30 | 0.32 | 0.33 | 0.59 | 0.63 | 0.20 | NA | 0.22 | 0.15 | 0.11 | 0.40 |
| | RMSE | 11.95 | 9.92 | 9.20 | 8.02 | 8.06 | 6.57 | 11.65 | 6.82 | 8.65 | 7.15 | 7.51 | 7.99 | 12.97 | 6.79 | 2.40 | NA | 2.78 | 1.92 | 1.41 | 5.04 |




Table 4. Annual mean absolute differences (ppb for gases and µg m⁻³ for particles) between the base case and the different emission perturbation scenarios as calculated by the different model groups over the European domain.

| Pollutant | Scenario | DE1 | DK1 | ES1 | FI1 | IT1 | IT2 | TR1 | UK1 | UK2 | FRES1 | All Mean | Common Mean |
|---|---|---|---|---|---|---|---|---|---|---|---|---|---|
| $O_3$ | GLO | -1.54 | -0.71 | | -0.40 | -0.37 | -0.63 | 2.83 | -0.83 | -0.79 | -0.63 | -0.34 | -0.82 |
| | NAM | -0.28 | -0.24 | 0.77 | -0.13 | | | -0.30 | -0.22 | | -0.22 | -0.09 | -0.22 |
| | EUR | -0.77 | 0.14 | | 0.09 | 0.43 | | | 0.06 | | 0.12 | 0.01 | -0.07 |
| $NO_2$ | GLO | -0.28 | -0.72 | | -1.20 | -0.93 | -0.95 | -1.93 | -0.75 | -1.10 | -0.89 | -0.97 | -0.77 |
| | NAM | 0.00 | 0.01 | 0.17 | 0.00 | 0.00 | | 0.01 | | | | 0.03 | 0.00 |
| | EUR | -0.30 | -0.69 | | -1.05 | -0.85 | | | -0.70 | | -0.89 | -0.75 | -0.73 |
| CO | GLO | -15.97 | -14.03 | | -21.10 | -18.13 | -15.04 | -26.01 | -12.83 | -16.94 | -16.11 | -17.35 | -16.01 |
| | NAM | -1.50 | -1.71 | 3.26 | -1.41 | | | -1.35 | -1.33 | | -1.55 | -0.80 | -1.50 |
| | EUR | -10.49 | -6.91 | | -14.63 | -10.11 | | | -7.87 | | -9.51 | -9.92 | -9.88 |
| $SO_2$ | GLO | -0.23 | -0.12 | | -0.17 | -0.17 | -0.11 | -0.23 | -0.20 | -0.28 | -0.15 | -0.18 | -0.17 |
| | NAM | 0.00 | 0.00 | 0.03 | 0.00 | | | 0.00 | 0.00 | | 0.00 | 0.00 | 0.00 |
| | EUR | -0.23 | -0.10 | | -0.14 | -0.13 | | | -0.16 | | -0.15 | -0.15 | -0.16 |
| $PM_{10}$ | GLO | -1.47 | -1.90 | | -2.52 | -2.97 | -1.58 | -3.58 | -2.32 | -2.81 | -2.27 | -2.38 | -2.10 |
| | NAM | -0.01 | -0.09 | 0.00 | -0.02 | | | -0.04 | -0.03 | | -0.04 | -0.03 | -0.04 |
| | EUR | -2.03 | -1.53 | | -2.20 | -2.46 | | | -1.96 | | -2.07 | -2.04 | -1.96 |
| $PM_{2.5}$ | GLO | -1.30 | -1.76 | | -2.15 | -2.56 | -1.33 | -2.79 | -1.78 | -2.44 | -2.10 | -2.02 | -1.82 |
| | NAM | 0.01 | -0.05 | 0.00 | -0.02 | | | -0.03 | -0.02 | | -0.04 | -0.02 | -0.02 |
| | EUR | -1.29 | -1.42 | | -1.82 | -2.05 | | | -1.47 | | -1.89 | -1.66 | -1.58 |






Table 5. Annual mean absolute differences (ppb for gases and µg m$^{-3}$ for particles) between the base case and the different emission perturbation scenarios as calculated by the different model groups over the North American domain.

| Pollutant | Scenario | DE1 | DK1 | US1 | US3 | All Mean | Common Mean |
|---|---|---|---|---|---|---|---|
| O$_3$ | GLO | -1.70 | -1.42 | -1.41 | -1.03 | -1.39 | -1.39 |
|  | NAM | -0.92 | -0.66 |  | -0.36 | -0.65 | -0.65 |
|  | EAS | -0.35 | -0.24 | -0.23 | -0.19 | -0.25 | -0.26 |
| NO$_2$ | GLO | -0.35 | -0.63 | -1.07 | -1.20 | -0.81 | -0.73 |
|  | NAM | -0.36 | -0.62 |  | -1.17 | -0.71 | -0.71 |
|  | EAS | 0.00 | 0.00 | 0.00 | -0.01 | 0.00 | 0.00 |
| CO | GLO | -9.31 | -20.48 | -22.12 | -25.01 | -19.23 | -18.27 |
|  | NAM | -3.84 | -13.35 |  | -19.87 | -12.35 | -12.35 |
|  | EAS | -2.60 | -4.16 | -3.64 | -3.07 | -3.37 | -3.28 |
| SO$_2$ | GLO | -0.33 | -0.32 | -0.48 | -0.25 | -0.34 | -0.30 |
|  | NAM | -0.33 | -0.32 |  | -0.48 | -0.37 | -0.37 |
|  | EAS | 0.00 | 0.00 |  | 0.00 | 0.00 | 0.00 |
| PM$_{10}$ | GLO | -2.26 | -0.66 |  | -4.24 | -2.39 | -2.39 |
|  | NAM | -2.02 | -0.59 |  | -4.19 | -2.27 | -2.27 |
|  | EAS | -0.56 | -0.05 |  | -0.03 | -0.21 | -0.21 |
| PM$_{2.5}$ | GLO | -0.60 | -1.67 |  | -2.29 | -1.52 | -1.52 |
|  | NAM | -0.62 | -1.56 |  | -2.24 | -1.47 | -1.47 |
|  | EAS | 0.01 | -0.04 |  | -0.03 | -0.02 | -0.02 |







Table 6. Annual mean RERER values calculated for the multi-model mean ensembles over Europe and North America.

|  | O$_3$ | NO$_2$ | CO | SO$_2$ | PM$_{10}$ | PM$_{2.5}$ |
|---|---|---|---|---|---|---|
| | | | EUROPE | | | |
| DE1 | 0.44 | -0.09 | 0.44 | 0.02 | 0.01 | 0.01 |
| DK1 | 0.85 | 0.23 | 0.63 | 0.37 | 0.17 | 0.28 |
| FI1 | 0.76 | -0.01 | 0.40 | 0.01 | 0.02 | 0.02 |
| FRES1 | 0.78 | 0.15 | 0.56 | 0.30 | 0.20 | 0.20 |
| IT1 | 1.10 | 0.34 | 0.93 | 0.42 | 0.27 | 0.26 |
| UK1 | 0.92 | 0.35 | 0.52 | 0.43 | 0.33 | 0.34 |
| MMM | 0.77 | 0.18 | 0.55 | 0.27 | 0.18 | 0.19 |
| | | | NORTH AMERICA | | | |
| DE1 | 0.77 | 0.12 | 0.73 | 0.07 | 0.09 | 0.12 |
| DK1 | 0.93 | 0.06 | 0.90 | 0.15 | 0.07 | 0.12 |
| US3 | 0.54 | 0.02 | 0.47 | 0.11 | 0.08 | 0.10 |
| MMM | 0.75 | 0.05 | 0.71 | 0.11 | 0.08 | 0.11 |





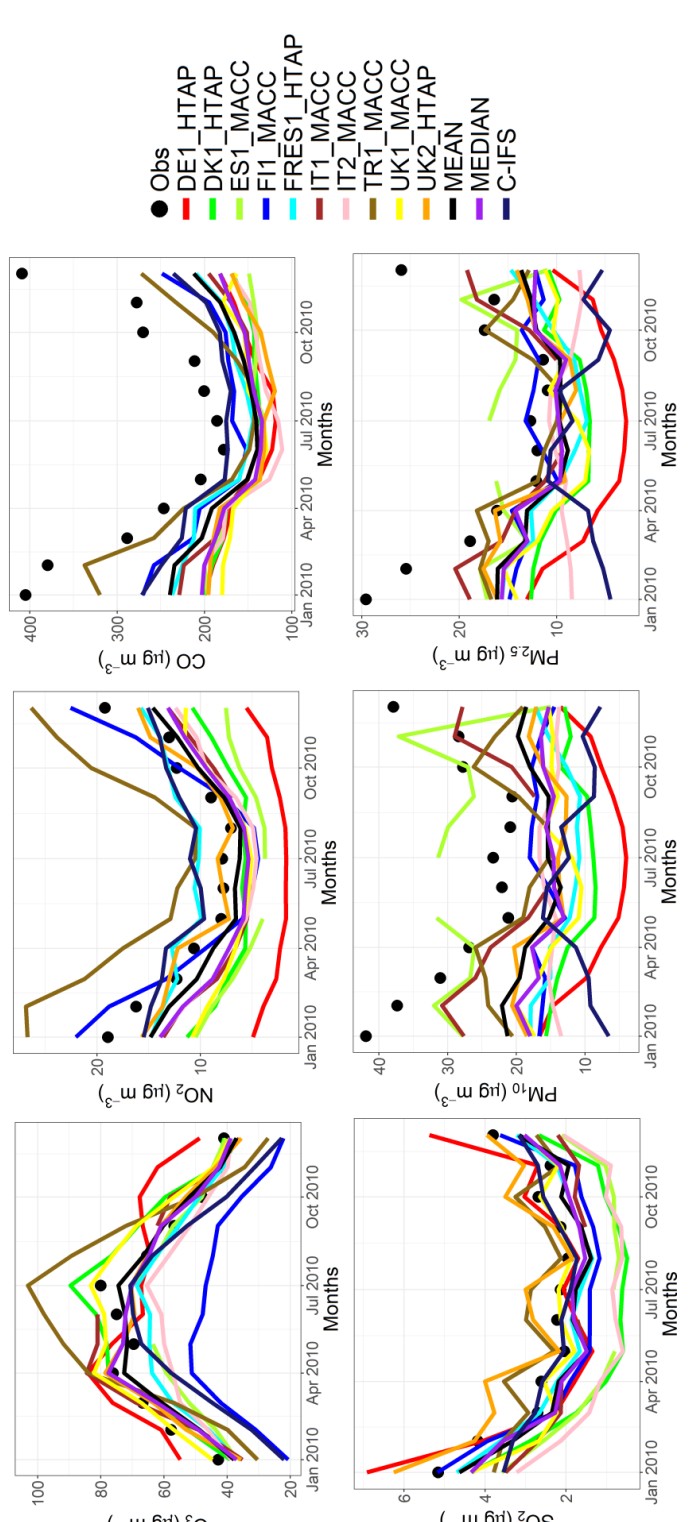


Fig.1. Observed and simulated monthly mean air pollutant levels, averaged over the monitoring stations over Europe.



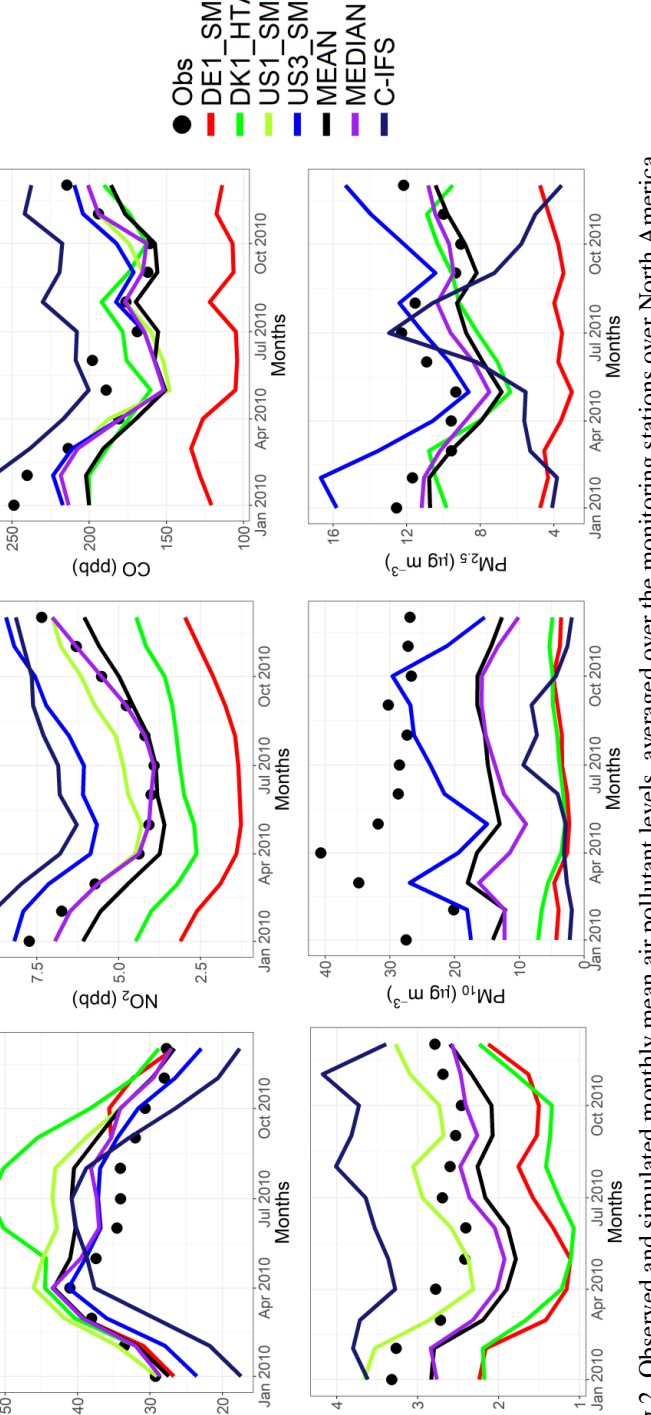

Fig.2. Observed and simulated monthly mean air pollutant levels, averaged over the monitoring stations over North America.





Fig.3. Multi-model mean air pollutant levels over Europe as simulated in the base case.







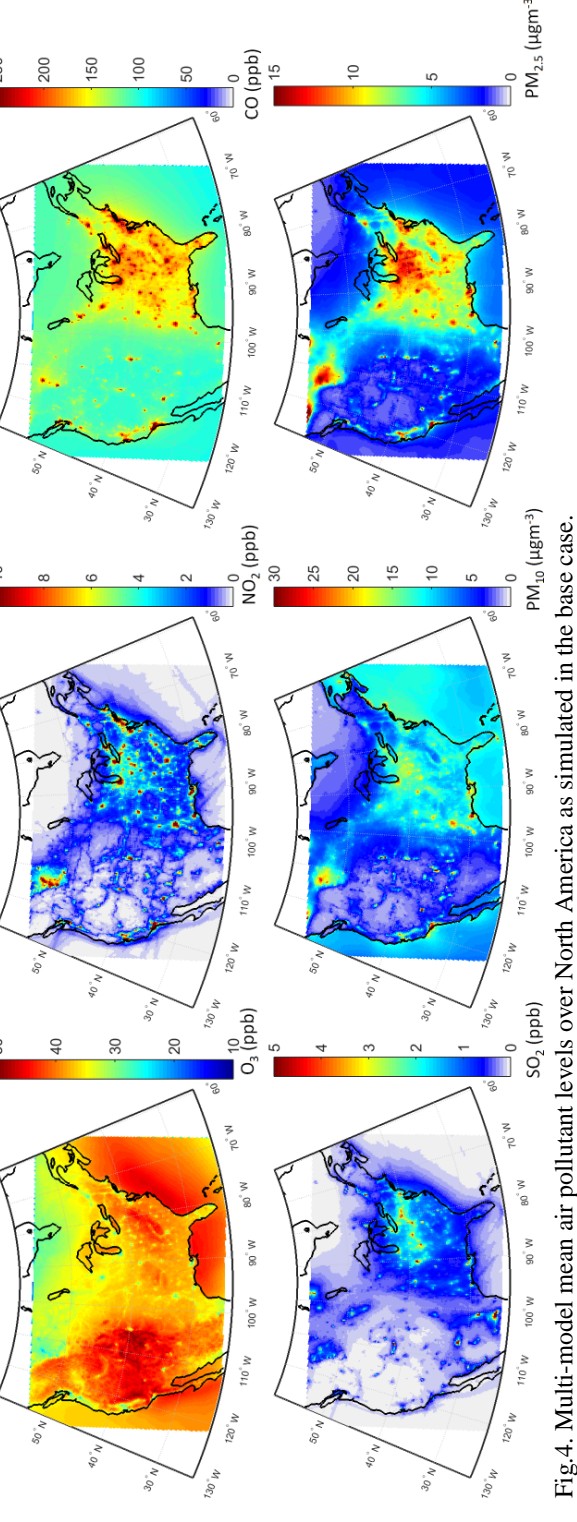

Fig. 4. Multi-model mean air pollutant levels over North America as simulated in the base case.



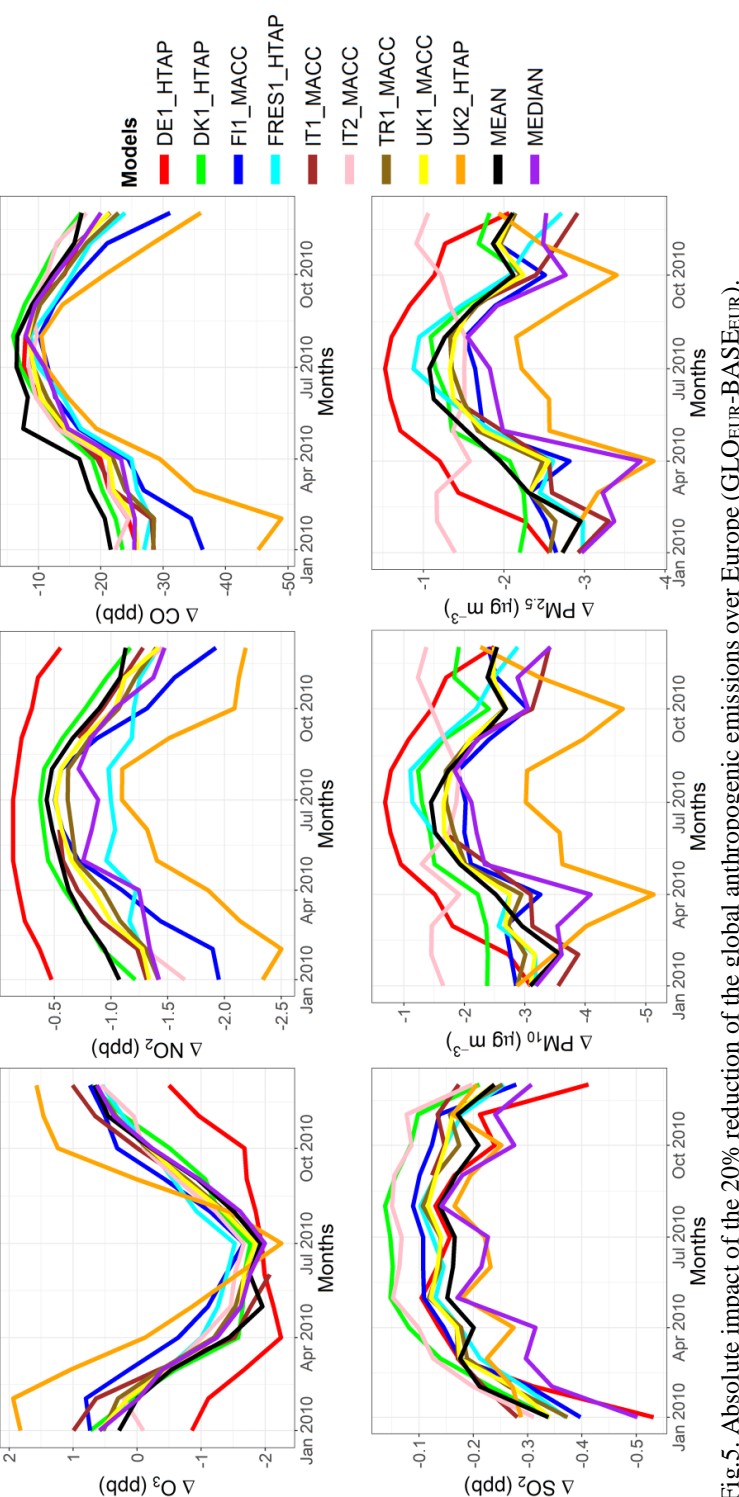

Fig.5. Absolute impact of the 20% reduction of the global anthropogenic emissions over Europe (GLO$_{EUR}$-BASE$_{EUR}$).






Fig.6. Spatial distribution of the annual mean relative differences between the global perturbation scenario and the base case over Europe as simulated by the multi-model mean ensemble.




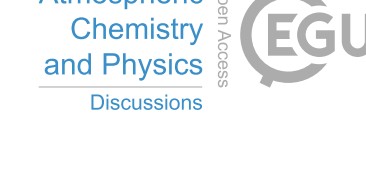

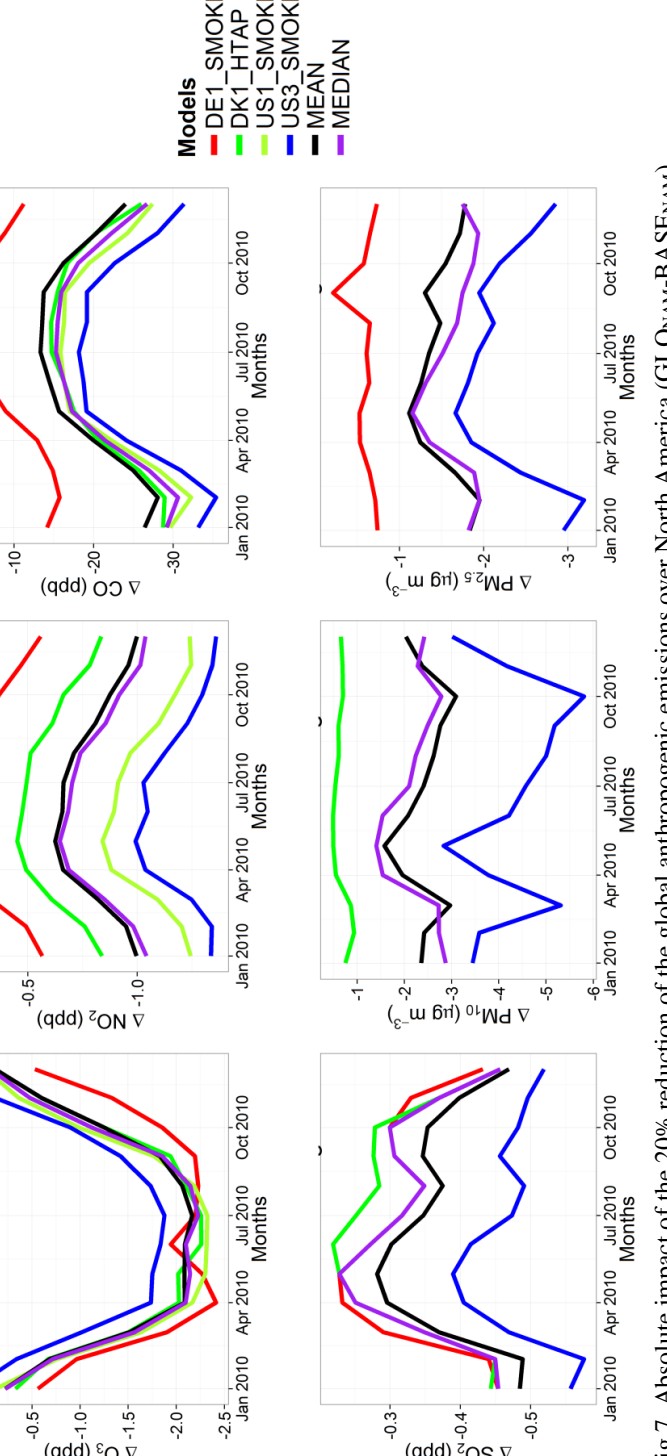

Fig.7. Absolute impact of the 20% reduction of the global anthropogenic emissions over North America (GLO$_{NAM}$-BASE$_{NAM}$).



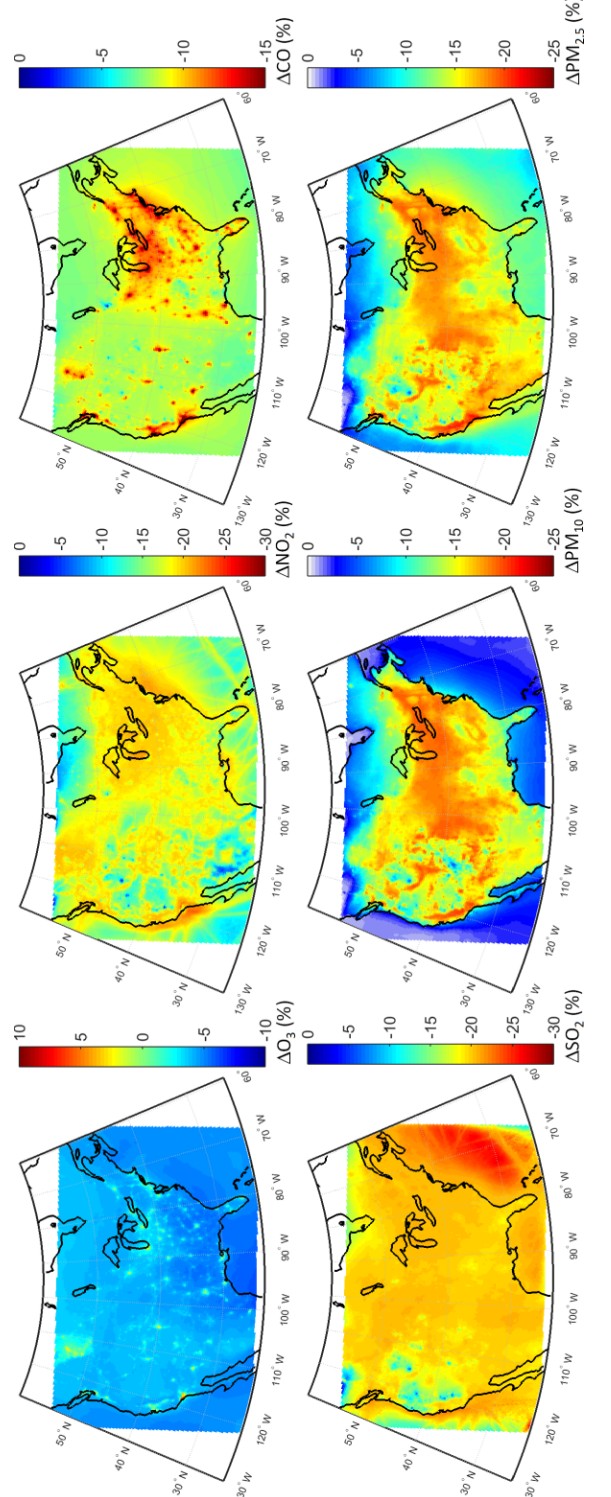

Fig.8. Spatial distribution of the annual mean relative differences between the global perturbation scenario and the base case over North America as simulated by the multi-model mean ensemble.






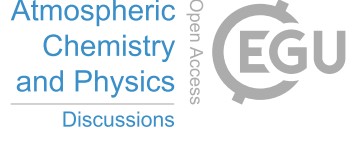

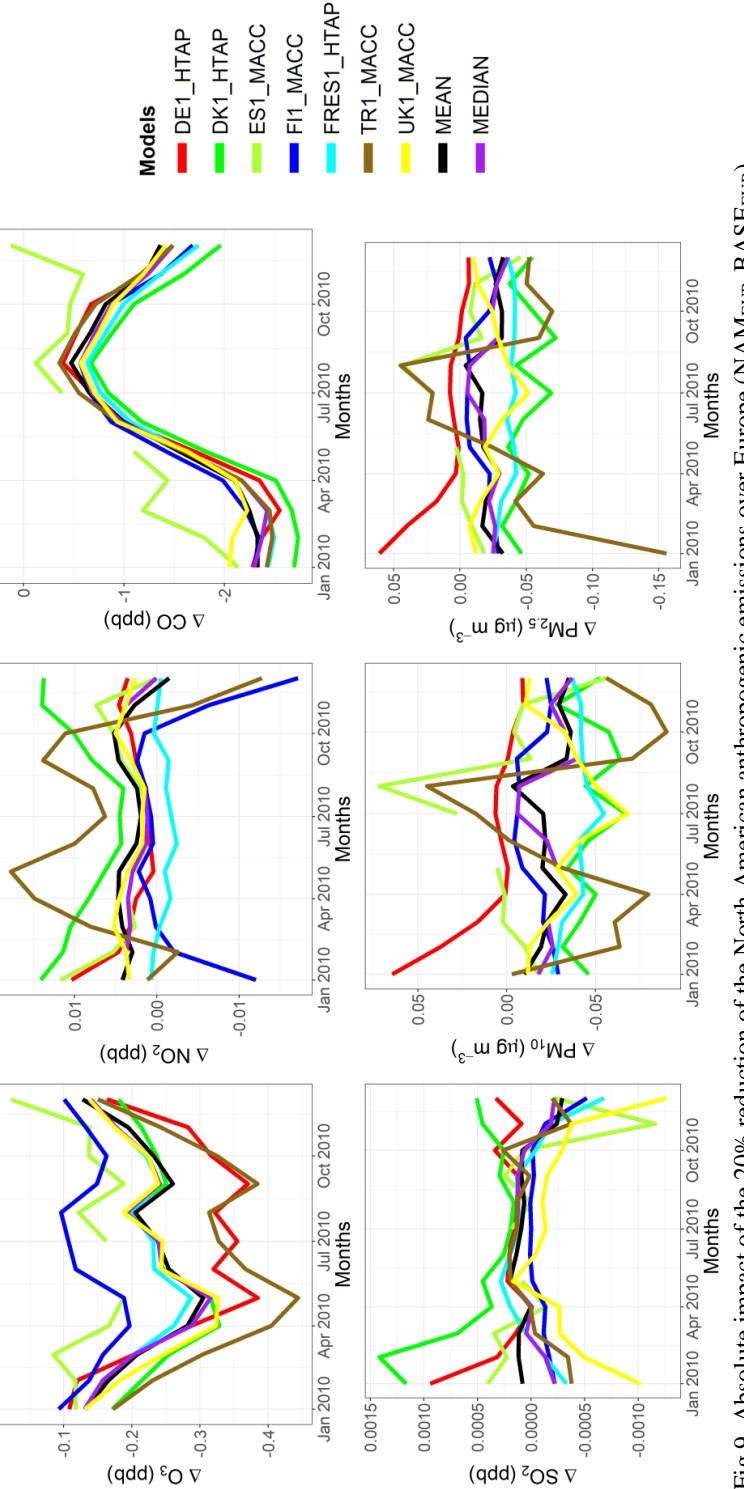

Fig.9. Absolute impact of the 20% reduction of the North American anthropogenic emissions over Europe ($NAM_{EUR}$-$BASE_{EUR}$).



Fig.10. Spatial distribution of the annual mean relative differences between the North American emissions perturbation scenario and the base case over Europe as simulated by the multi-model mean ensemble.






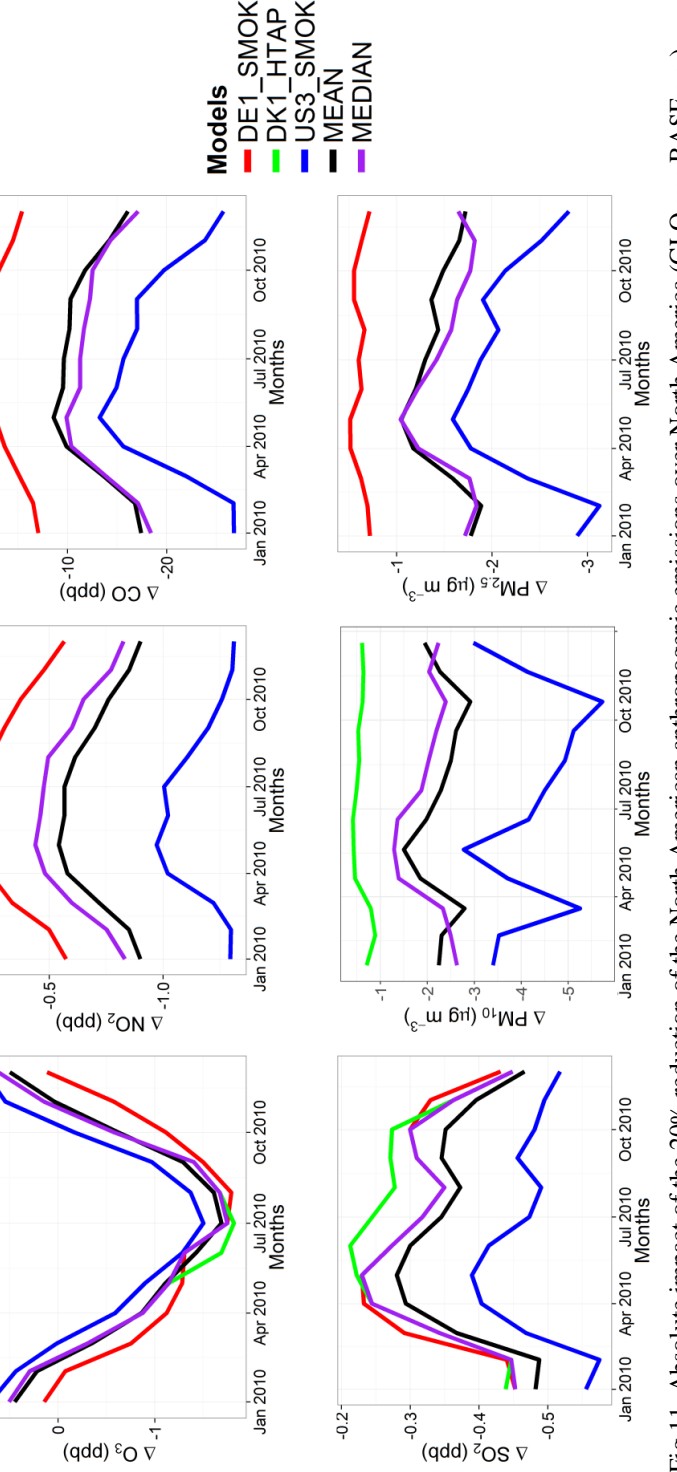

Fig.11. Absolute impact of the 20% reduction of the North American anthropogenic emissions over North America (GLO$_{\mathrm{NAM}}$-BASE$_{\mathrm{NAM}}$).






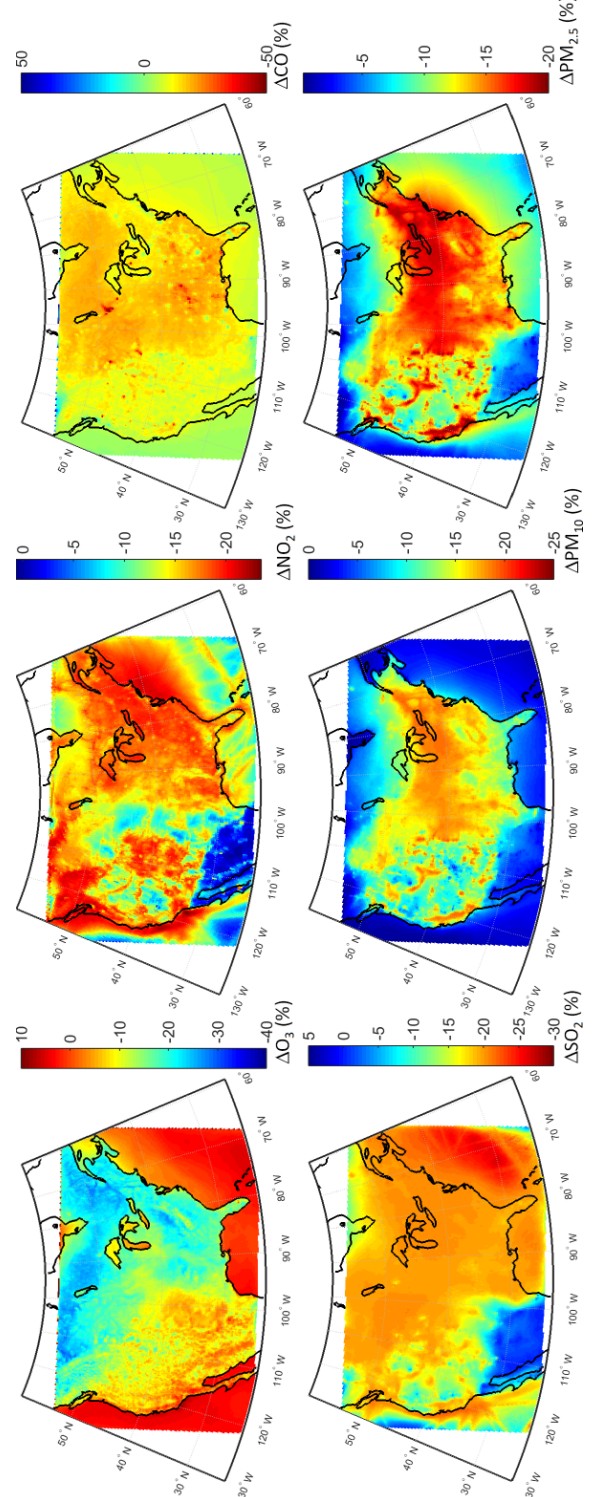

Fig.12. Spatial distribution of the annual mean relative differences between the North American emissions perturbation scenario and the base case over North America as simulated by the multi-model mean ensemble.







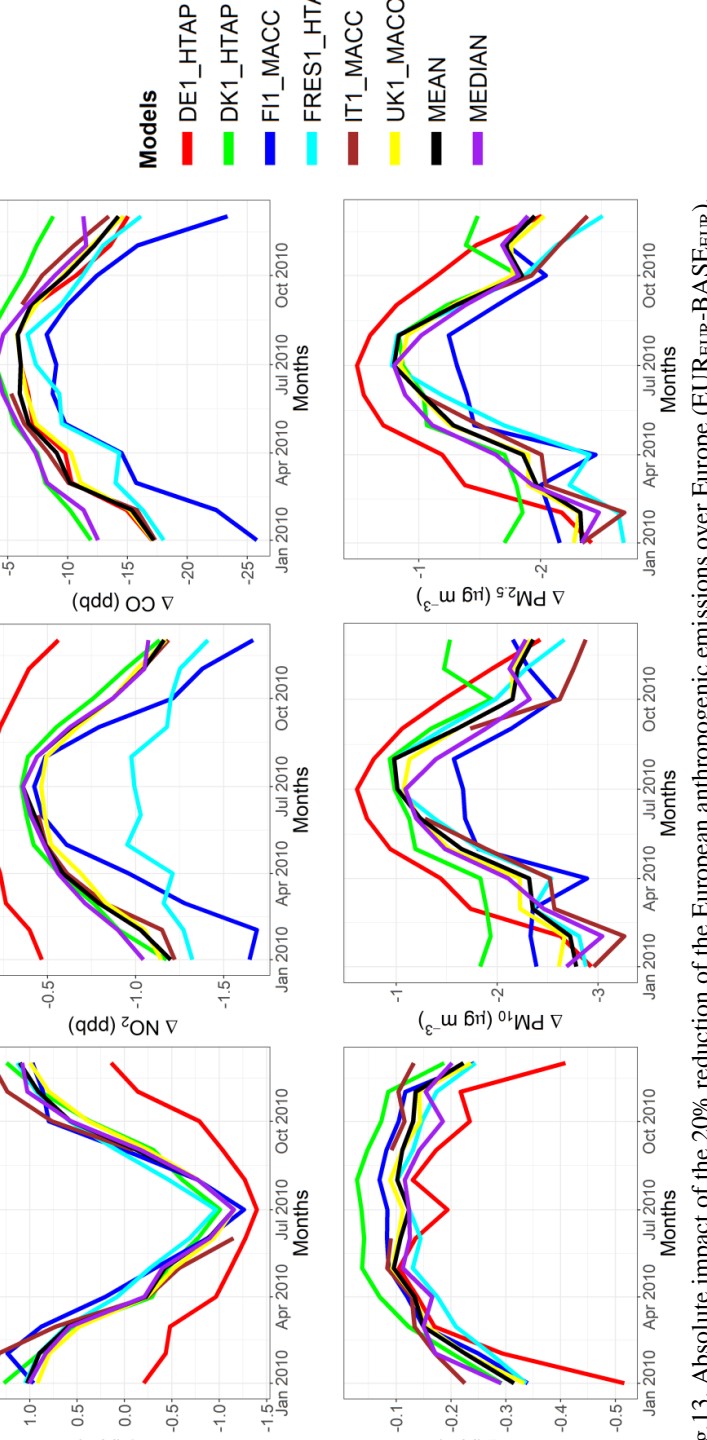

Fig.13. Absolute impact of the 20% reduction of the European anthropogenic emissions over Europe ($EUR_{EUR}$-$BASE_{EUR}$).







Fig.14. Spatial distribution of the annual mean relative differences between the European emissions perturbation scenario and the base case over Europe as simulated by the multi-model mean ensemble.




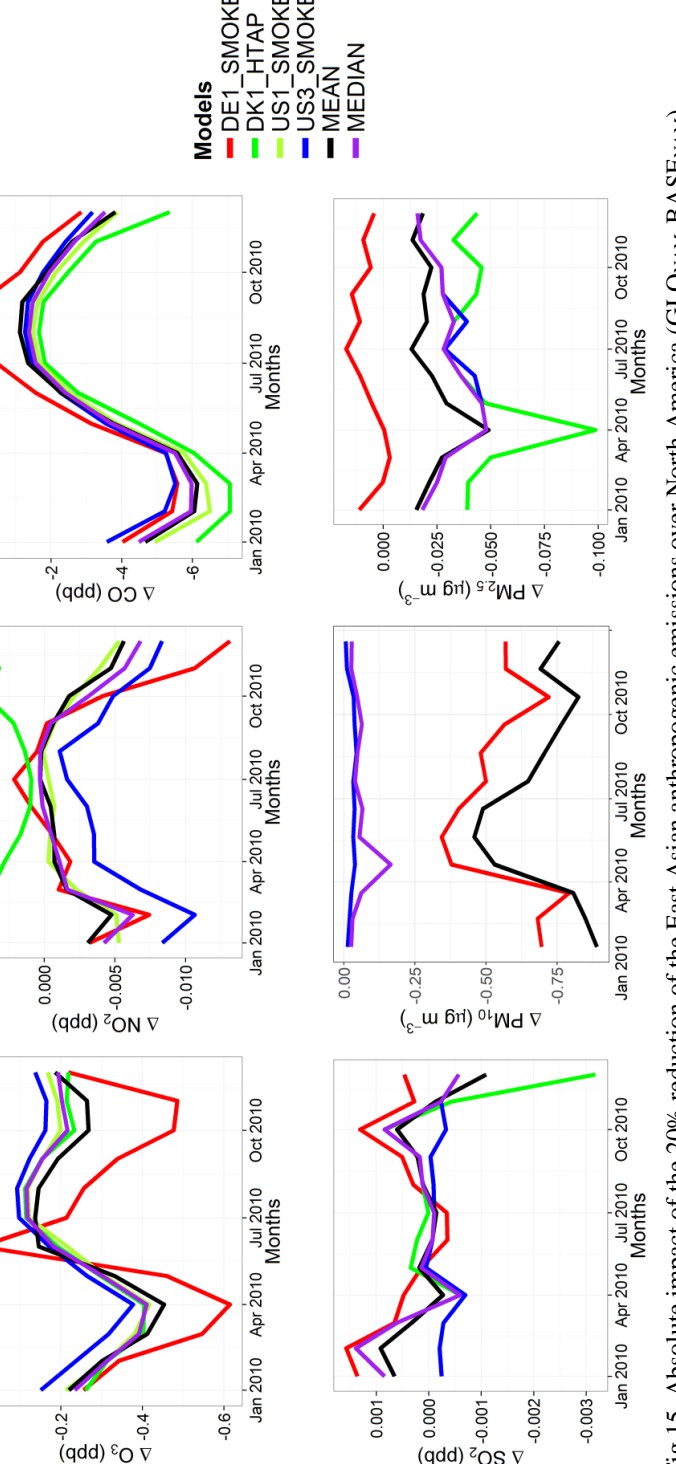

Fig.15. Absolute impact of the 20% reduction of the East Asian anthropogenic emissions over North America (GLO$_{NAM}$-BASE$_{NAM}$).




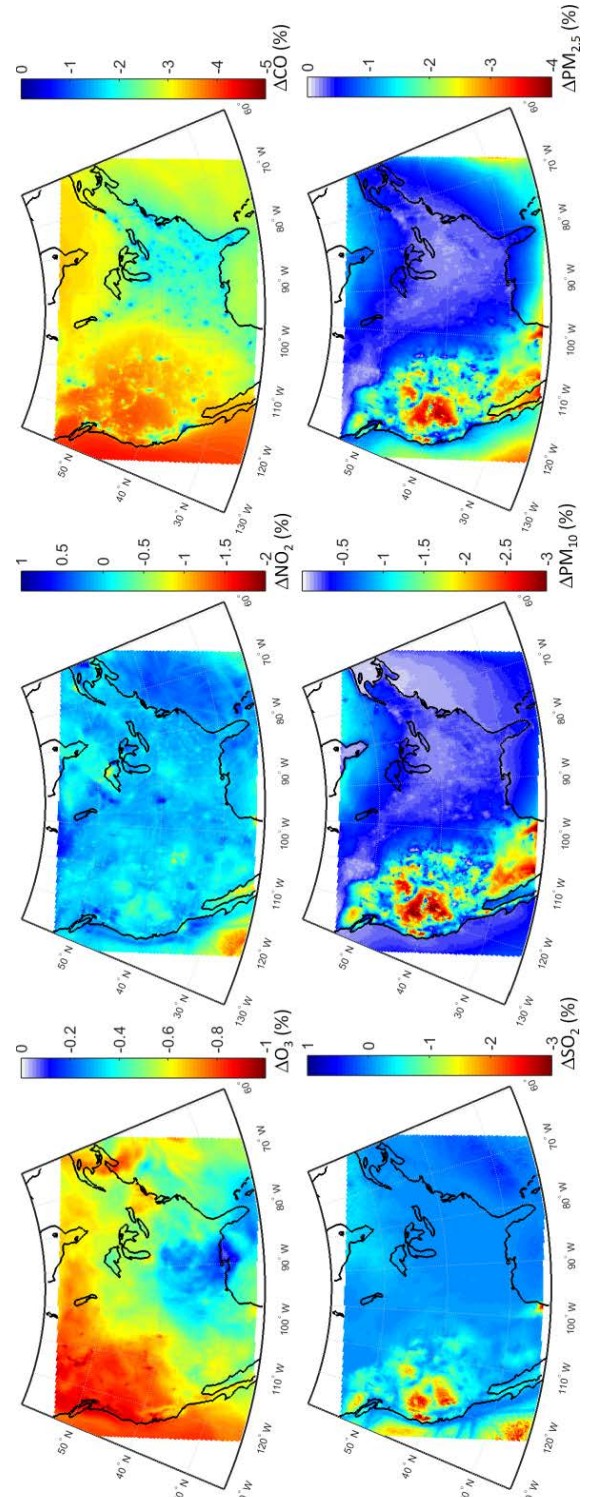

Fig.16. Spatial distribution of the annual mean relative differences between the East Asian emissions perturbation scenario and the base case over North America as simulated by the multi-model mean ensemble.

903

904

905