# Peer review of "Influence of anthropogenic emissions and boundary conditions on multi-model"

_Atmospheric Chemistry and Physics, 2017_

## Referee Comment (RC1) · Anonymous Referee #1 · 10 Apr 2018

Im et al. present a multi-model assessment of the impacts due to reductions in domestic and global emissions on O3, NO2, CO, SO2, PM10 and PM2.5 over the North America and Europe in the framework of the Air Quality Model Evaluation International Initiative -phase 3 (AQMEII3). Quantitative information using results from an ensemble of models is provided and the manuscript is recommended for publication in ACP.

Few suggestions are listed below for authors to consider during the revision.

Section 3.1: Further discussions are desirable describing (and attributing) the model biases in terms of setup, errors in model meteorology, and processes included /excluded in different simulations.

WRF simulated meteorology shows large spatial variability across Europe with lower correlations over Alps and coasts (e.g. Mar et al., GMD, 2016). It should be discussed whether the strong model biases in some of the simulations shown here (e. g. TR1_MACC for O3 in Fig. 1) are arising from only few specific areas or that models are biased in general over the entire region.

Page 10, l. 393-395: Why does SO2 enhancement in case of reduced domestic emissions in North America are pronounced in a small belt over Europe? Is it possible to substantiate the statements with model simulated OH fields?

Page 4, l.162 – "where embedded" to "were embedded"

Page 7, l.276: "SO2….by 35% 5". Pl. check this sentence.

Page 7, l.279 – "effect" to "affects"

---

## Referee Comment (RC2) · Anonymous Referee #2 · 17 Apr 2018

The manuscript describes results from AQMEII3 exercises studying the impact of global and regional emission reductions using regional models fed by the global model C-IFS.

There is a lot of work behind model ensembles and inter-comparisons. The manuscript present results mainly in a clear manner, but there are sections that need to be clarified. I also see that information about the model simulations included is lacking. I have the

following main points that I think important for the authors to address before publication of the manuscript in ACP:

- Table 1. You are ordering runs according to groups, not according to models. After a more thorough look many of the groups use the same model, sometimes even on the same resolution. What is the use of an ensemble of groups running the same model? Ideally this should give exactly the same results unless someone makes an error or the model version is different.

- Linked to the previous bullet: you end the conclusions with raising the issue of the impact of different model parameterization. However, you do not include such information. You should include a description of important model facts and add a discussion on these linking them to your results.

- Based on this information, perhaps some model runs should be removed from the ensemble (too many of the same model? Too simple parameterizations for some species?).

- In the abstract you describe daily maximum 8h mean ozone. Is this what you show and evaluate in the tables and figures? Or is it monthly/annual means? You need to clarify this (in all figures/table legends as well as in the methods) or (/and) only include results in the abstract which you are actually showing as results in figures/tables.

- The RERER value analysis is interesting. It would be of great value if you describe the ozone RERER value based on monthly values (daily max 8h mean or mean), since ozone formation capacity/local contribution is seasonally dependent. Perhaps you can come up with a smart way of illustrating these rather than just adding more table values.

- I don't see the point of showing figures 11 (GLONAM-BASENAM) to 14 (EUREUR-BASEEUR). I would much rather see geographically resolved RERER values as a complement to the other figures.

- Line 221-224. The method of first taking difference then calculating mean is only valid

[Figure]

if you are working with means. How do you treat the daily maximum 8h mean? Is the method valid for this metric (if that is what you are showing in the figures for ozone).

- Table 1. The number of simulations (scenarios) is different when comparing the table to the method text (for Europe). An x is missing in the table (grey area for north America-region).

- Table 3. You state unit: % for NMB and NMGE, but the values in the table are clearly without unit. You should not have different units for North America and Europe (for RMSE in this case).

- You have a supplement but you do not refer to it in your manuscript.

- The figure legend of S3 is incorrect.

- Section 2, first paragraph is messy and repetitive.

---

## Author Comment (AC1) · 11 Jun 2018

We would like to thank the reviewer for the positive feedback. Below, we reply to the comments from the reviewer:

Comment 1: Section 3.1: Further discussions are desirable describing (and attributing) the model biases in terms of setup, errors in model meteorology, and processes included /excluded in different simulations. Response: We have now included more detailed model descriptions regarding the chemistry and aerosol modules in the Mate-

[Figure]

rials and Methods section (Lines 152-170), along with an updated and extended Table 1 providing references to chemical mechanisms used in the models. In addition, we have also provided more discussion on possible model biases in section 3.1 (Lines 280-292), including some discussion on the meteorological biases. However, this paper does not aim to make a full evaluation and error attribution of the models. It does however build on Solazzo et al. (2017) in the same special issue that makes a deep evaluation of the models.

Comment: Page 10, l. 393-395: Why does $SO_2$ enhancement in case of reduced domestic emissions in North America are pronounced in a small belt over Europe? Is it possible to substantiate the statements with model simulated OH fields? Response: We thank the reviewer for the careful review and we have identified a problem during the plotting. We have now corrected this plot. However, there is still a slight increase of $SO_2$ over the Alps that is simulated by the majority of the models. The AQMEII database unfortunately does not include OH fields so we cannot further evaluate this increase in this paper.

Comment: Page 4, l.162 – "where embedded" to "were embedded" Response: Corrected (Line 184).

Comment: Page 7, l.276: "$SO_2$: : :.by 35% 5". Pl. check this sentence. Response: Corrected (Line 305).

Comment: Page 7, l.279 – "effect" to "affects" Response: Corrected (Line 308).

---

## Author Comment (AC2) · 11 Jun 2018

We would like to thank the reviewer for the careful read of the manuscript positive feedback. Below, we reply to the comments from the reviewer:

Comment: - Table 1. You are ordering runs according to groups, not according to models. After a more thorough look many of the groups use the same model, sometimes even on the same resolution. What is the use of an ensemble of groups running the same model? Ideally this should give exactly the same results unless someone makes

[Figure]

an error or the model version is different. - Linked to the previous bullet: you end the conclusions with raising the issue of the impact of different model parameterization. However, you do not include such information. You should include a description of important model facts and add a discussion on these linking them to your results. - Based on this information, perhaps some model runs should be removed from the ensemble (too many of the same model? Too simple parameterizations for some species?). Response: We have now updated and extended Table 1, providing more information on the mode specific spatial and vertical resolutions as well different chemistry and aerosol mechanisms. We have also added more information on the differences between the versions of the same models (e.g. CMAQ and WRF-Chem) by each group (Lines 152-170). The models or the versions of the same models differ from each and therefore, we think model removal is not necessary.

Comment: In the abstract you describe daily maximum 8h mean ozone. Is this what you show and evaluate in the tables and figures? Or is it monthly/annual means? You need to clarify this (in all figures/table legends as well as in the methods) or (/and) only include results in the abstract which you are actually showing as results in figures/tables. Response: The model evaluation is based on monthly means, as described in the beginning of section 3.1. and the cation of Table 3 and the captions of Fig. 1 and 2. We have calculated the impact on daily maximum 8hr O3 in order to show a policy impact of these reductions.

Comment: The RERER value analysis is interesting. It would be of great value if you describe the ozone RERER value based on monthly values (daily max 8h mean or mean), since ozone formation capacity/local contribution is seasonally dependent. Perhaps you can come up with a smart way of illustrating these rather than just adding more table values. - I don't see the point of showing figures 11 (GLONAM-BASENAM) to 14 (EUREURBASEEUR). I would much rather see geographically resolved RERER values as a complement to the other figures. Response: We thank the reviewer for his interest in the RERER analyses and we agree that is can be more emphasized in

the paper. Therefore, we have now, as suggested by the reviewer, produced spatial distribution maps for O3 and PM2.5 (Fig. 17) as well as monthly time series of the response for these pollutants (Fig 18) and added discussions on these results (Lines 578-606). On the other hand, we would like to keep Figs 11 and 14 to be consistent in the flow.

Comment: Line 221-224. The method of first taking difference then calculating mean is only valid if you are working with means. How do you treat the daily maximum 8h mean? Is the method valid for this metric (if that is what you are showing in the figures for ozone). Response: The figures and tables only show the differences in monthly and annual means of the pollutants. Daily maximum 8hr ozone is only presented in the text as an additional information. As written in the text, we look at the difference in the mean of daily maximum 8hr ozone, but these are not presented in tables or figures.

Comment: Table 1. The number of simulations (scenarios) is different when comparing the table to the method text (for Europe). An x is missing in the table (grey area for north America-region). Response: We thank the reviewer for the careful read. We have now corrected these.

Comment: Table 3. You state unit: % for NMB and NMGE, but the values in the table are clearly without unit. You should not have different units for North America and Europe (for RMSE in this case). Response: We agree with the reviewer and we have now corrected the units in Table 3 caption.

Comment: You have a supplement but you do not refer to it in your manuscript. Response: We thank the reviewer for pointing out this missing part. We have now referred to the supplement in various parts of the manuscript (Lines 194, 418-419, 520-521).

Comment: The figure legend of S3 is incorrect. Response: We have now corrected the figure caption.

Comment: Section 2, first paragraph is messy and repetitive. Response: We have now

reorganized this paragraph (Lines 171-190).